# MagicDrive: Street View Generation with Diverse 3D Geometry Control

**Ruiyuan Gao**[1*]**, Kai Chen**[2*]**, Enze Xie**[3†]**, Lanqing Hong**[3]**,**
**Zhenguo Li**[3]**, Dit-Yan Yeung**[2]**, Qiang Xu**[1†]
[1]The Chinese University of Hong Kong  [2]Hong Kong University of Science and Technology
[3]Huawei Noah's Ark Lab
{rygao,qxu}@cse.cuhk.edu.hk, kai.chen@connect.ust.hk,
{xie.enze,honglanqing,li.zhenguo}@huawei.com, dyyeung@cse.ust.hk

## Abstract

Recent advancements in diffusion models have significantly enhanced the data synthesis with 2D control. Yet, precise 3D control in street view generation, crucial for 3D perception tasks, remains elusive. Specifically, utilizing Bird's-Eye View (BEV) as the primary condition often leads to challenges in geometry control (*e.g.*, height), affecting the representation of object shapes, occlusion patterns, and road surface elevations, all of which are essential to perception data synthesis, especially for 3D object detection tasks. In this paper, we introduce MagicDrive, a novel street view generation framework, offering diverse 3D geometry controls including camera poses, road maps, and 3D bounding boxes, together with textual descriptions, achieved through tailored encoding strategies. Besides, our design incorporates a cross-view attention module, ensuring consistency across multiple camera views. With MagicDrive, we achieve high-fidelity street-view image & video synthesis that captures nuanced 3D geometry and various scene descriptions, enhancing tasks like BEV segmentation and 3D object detection.

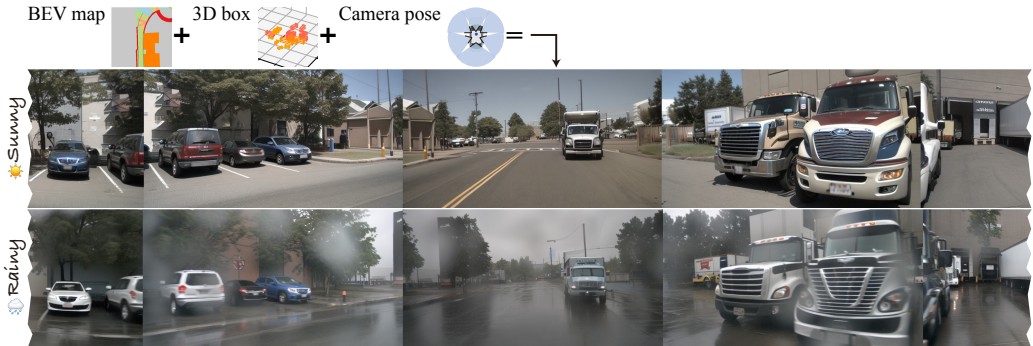

Figure 1: Multi-camera street view generation from MagicDrive. MagicDrive can generate continuous camera views with controls from the road map, object boxes, and text (*e.g.*, weather).

## 1 Introduction

The high costs associated with data collection and annotation often impede the effective training of deep learning models. Fortunately, cutting-edge generative models have illustrated that synthetic data can notably boost performance across various tasks, such as object detection (Chen et al., 2023c) and semantic segmentation (Wu et al., 2023b). Yet, the prevailing methodologies are largely tailored to 2D contexts, primarily relying on 2D bounding boxes (Lin et al., 2014; Han et al., 2021) or segmentation maps (Zhou et al., 2019) as layout conditions (Chen et al., 2023c; Li et al., 2023b).

In autonomous driving applications, a thorough grasp of the 3D environment is essential. This demands reliable techniques for tasks like Bird's-Eye View (BEV) map segmentation (Zhou & Krähenbühl, 2022; Ji et al., 2023) and 3D object detection (Chen et al., 2020; Huang et al., 2021; Liu et al., 2023a; Ge et al., 2023). A genuine 3D geometry representation is crucial for capturing intricate details from 3D annotations, such as road elevations, object heights, and their occlusion pat-

---

*Equal contribution. †Corresponding authors. Project Page: https://flymin.github.io/magicdrive.

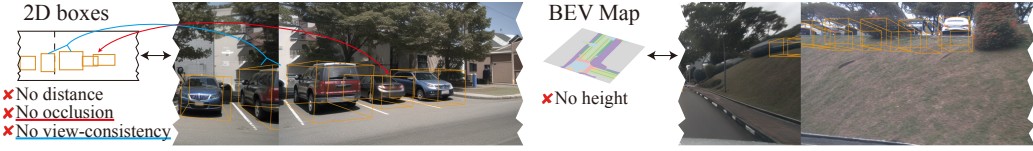

(a) 3D bounding boxes show position relationships.  (b) Road surface elevation guided by 3D bounding boxes.

Figure 2: 3D bounding boxes are crucial for street view synthesis. Two examples show that 2D boxes or BEV maps lost distance, height, and elevation. Images are generated from MAGICDRIVE.

terns, as shown in Figure 2. Consequently, generating multi-camera street-view images according to 3D annotations becomes vital to boost downstream perception tasks.

For street-view data synthesis, two pivotal criteria are *realism* and *controllability*. *Realism* requires that the quality of the synthetic data should align with that of real data; and in a given scene, views from varying camera perspectives should remain consistent with one another (Mildenhall et al., 2020). On the other hand, *controllability* emphasizes the precision in generating street-view images that adhere to provided conditions: the BEV map, 3D object bounding boxes, and camera poses for views. Beyond these core requirements, effective data augmentation should also grant the flexibility to tweak finer scenario attributes, such as prevailing weather conditions or the time of day. Existing solutions like BEVGen (Swerdlow et al., 2023) approach street view generation by encapsulating all semantics within BEV. Conversely, BEVControl (Yang et al., 2023a) starts by projecting 3D coordinates to image views, subsequently using 2D geometric guidance. However, both methods compromise certain geometric dimensions—height is lost in BEVGen and depth in BEVControl.

The rise of diffusion models has significantly pushed the boundaries of controllable image generation quality. Specifically, ControlNet (Zhang et al., 2023a) proposes a flexible framework to incorporate 2D spatial controls based on pre-trained Text-to-Image (T2I) diffusion models (Rombach et al., 2022). However, 3D conditions are distinct from pixel-level conditions or text. The challenge of seamlessly integrating them with multi-camera view consistency in street view synthesis remains.

In this paper, we introduce MAGICDRIVE, a novel framework dedicated to street-view synthesis with diverse 3D geometry controls[1]. For *realism*, we harness the power of pre-trained stable diffusion (Rombach et al., 2022), further fine-tuning it for street view generation. One distinctive component of our framework is the cross-view attention module. This simple yet effective component provides multi-view consistency through interactions between adjacent views. In contrast to previous methods, MAGICDRIVE proposes a separate design for objects and road map encoding to improve *controllability* with 3D data. More specifically, given the sequence-like, variable-length nature of 3D bounding boxes, we employ cross-attention akin to text embeddings for their encoding. Besides, we propose that an addictive encoder branch like ControlNet (Zhang et al., 2023a) can encode maps in BEV and is capable of view transformation. Therefore, our design achieves geometric controls without resorting to any explicit geometric transformations or imposing geometric constraints on multi-camera consistency. Finally, MAGICDRIVE factors in textual descriptions, offering attribute control such as weather conditions and time of day.

Our MAGICDRIVE framework, despite its simplicity, excels in generating strikingly realistic images & videos that align with road maps, 3D bounding boxes, and varied camera perspectives. Besides, the images produced can enhance the training for both 3D object detection and BEV segmentation tasks. Furthermore, MAGICDRIVE offers comprehensive geometric controls at the *scene*, *background*, and *foreground* levels. This flexibility makes it possible to craft previously unseen street views suitable for simulation purposes. We summarize the main contributions of this work as:

- The introduction of MAGICDRIVE, an innovative framework that generates multi-perspective camera views & videos conditioned on BEV and 3D data tailored for autonomous driving.

- The development of simple yet potent strategies to manage 3D geometric data, effectively addressing the challenges of multi-camera view consistency in street view generation.

- Through rigorous experiments, we demonstrate that MAGICDRIVE outperforms prior street view generation techniques, notably for the multi-dimensional controllability. Additionally, our results reveal that synthetic data delivers considerable improvements in 3D perception tasks.

---

[1]In this paper, our 3D geometry controls contain control from road maps, 3D object boxes, and camera poses. We do not consider others like the exact shape of objects or background contents.

## 2 RELATED WORK

**Diffusion Models for Conditional Generation.** Diffusion models (Ho et al., 2020; Song et al., 2020; Zheng et al., 2023) generate images by learning a progressive denoising process from the Gaussian noise distribution to the image distribution. These models have proven exceptional across diverse tasks, such as text-to-image synthesis (Rombach et al., 2022; Nichol et al., 2022; Yang et al., 2023b), inpainting (Wang et al., 2023a), and instructional image editing (Zhang et al., 2023b; Brooks et al., 2023), due to their adaptability and competence in managing various form of controls (Zhang et al., 2023a; Li et al., 2023b) and multiple conditions (Liu et al., 2022a; Gao et al., 2023). Besides, data synthesized from geometric annotations can aid downstream tasks such as 2D object detection (Chen et al., 2023c; Wu et al., 2023b). Thus, this paper explores the potential of T2I diffusion models in generating street-view images and benefiting downstream 3D perception models.

**Street View Generation.** Numerous street view generation models condition on 2D layouts, such as 2D bounding boxes (Li et al., 2023b) and semantic segmentation (Wang et al., 2022). These methods leverage 2D layout information corresponding directly to image scale, whereas the 3D information does not possess this property, thereby rendering such methods unsuitable for leveraging 3D information for generation. For street view synthesis with 3D geometry, BEVGen (Swerdlow et al., 2023) is the first to explore. It utilizes a BEV map as a condition for both roads and vehicles. However, the omission of height information limits its application in 3D object detection. BEVControl (Yang et al., 2023a) amends the loss of object's height by the height-lifting process. Similarly, Wang et al. (2023b) also projects 3D boxes to camera views to guide generation. However, the projection from 3D to 2D results in the loss of essential 3D geometric information, like depth and occlusion. In this paper, we propose to encode bounding boxes and road maps separately for more nuanced control and integrate scene descriptions, offering enhanced control over the generation of street views.

**Multi-camera Image Generation** of a 3D scene fundamentally requires viewpoint consistency. Several studies have addressed this issue within the context of indoor scenes. For instance, MVDiffusion (Tang et al., 2023) employs panoramic images and a cross-view attention module to maintain global consistency, while Tseng et al. (2023) leverage epipolar geometry as a constraining prior. These approaches, however, primarily rely on the continuity of image views, a condition not always met in street views due to limited camera overlap and different camera configurations (*e.g.*, exposure, intrinsic). Our MAGICDRIVE introduces extra cross-view attention modules to UNet, which significantly enhances consistency across multi-camera views.

## 3 PRELIMINARY

**Problem Formulation.** In this paper, we consider the coordinate of the LiDAR system as the ego car's coordinate, and parameterize all geometric information according to it. Let $\mathbf{S} = \{\mathbf{M}, \mathbf{B}, \mathbf{L}\}$ be the description of a driving scene around the ego vehicle, where $\mathbf{M} \in \{0,1\}^{w \times h \times c}$ is the binary map representing a $w \times h$ meter road area in BEV with $c$ semantic classes, $\mathbf{B} = \{(c_i, b_i)\}_{i=1}^{N}$ represents the 3D bounding box position ($b_i = \{(x_j, y_j, z_j)\}_{j=1}^{8} \in \mathbb{R}^{8 \times 3}$) and class ($c_i \in \mathcal{C}$) for each object in the scene, and $\mathbf{L}$ is the text describing additional information about the scene (*e.g.*, weather and time of day). Given a camera pose $\mathbf{P} = [\mathbf{K}, \mathbf{R}, \mathbf{T}]$ (*i.e.*, intrinsics, rotation, and translation), the goal of street-view image generation is to learn a generator $\mathcal{G}(\cdot)$ which synthesizes realistic images $I \in \mathbb{R}^{H \times W \times 3}$ corresponding to the scene $\mathbf{S}$ and camera pose $\mathbf{P}$ as, $I = \mathcal{G}(\mathbf{S}, \mathbf{P}, z)$, where $z \sim \mathcal{N}(0, 1)$ is a random noise from Gaussian distribution.

**Conditional Diffusion Models.** Diffusion models (Ho et al., 2020; Song et al., 2020) generate data ($\boldsymbol{x}_0$) by iteratively denoising a random Gaussian noise ($\boldsymbol{x}_T$) for $T$ steps. Typically, to learn the denoising process, the network is trained to predict the noise by minimizing the mean-square error:

$$\ell_{simple} = \mathbb{E}_{\boldsymbol{x}_0, \boldsymbol{c}, \boldsymbol{\epsilon}, t} \left[ ||\boldsymbol{\epsilon} - \boldsymbol{\epsilon}_\theta(\sqrt{\bar{\alpha}_t} \boldsymbol{x}_0 + \sqrt{1 - \bar{\alpha}_t} \boldsymbol{\epsilon}, t, \boldsymbol{c})||^2 \right], \tag{1}$$

where $\boldsymbol{\epsilon}_\theta$ is the network to train, with parameters $\theta$, $\boldsymbol{c}$ is optional conditions, which is used for the conditional generation, $t \in [0, T]$ is the time-step, $\boldsymbol{\epsilon} \in \mathcal{N}(0, I)$ is the additive Gaussian noise, and $\bar{\alpha}_t$ is a scalar parameter. Latent diffusion models (LDM) (Rombach et al., 2022) is a special kind of diffusion model, where they utilize a pre-trained Vector Quantized Variational AutoEncoder (VQ-VAE) (Esser et al., 2021) and perform diffusion process in the latent space. Given the VQ-VAE encoder as $z = \mathcal{E}(x)$, one can rewrite $\boldsymbol{\epsilon}_\theta(\cdot)$ in Equation 1 as $\boldsymbol{\epsilon}_\theta(\sqrt{\bar{\alpha}_t}\mathcal{E}(\boldsymbol{x}_0) + \sqrt{1 - \bar{\alpha}_t}\boldsymbol{\epsilon}, t, \boldsymbol{c})$ for LDM. Besides, LDM considers text describing the image as condition $c$.

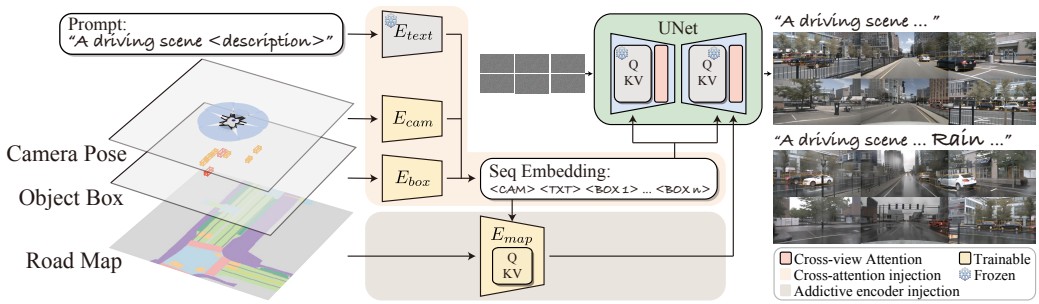

Figure 3: Overview of MAGICDRIVE for street-view image generation. MAGICDRIVE generates highly realistic images, exploiting geometric information from 3D annotations by independently encoding road maps, object boxes, and camera parameters for precise, geometry-guided synthesis. Additionally, MAGICDRIVE accommodates guidance from descriptive conditions (*e.g.*, weather).

## 4 STREET VIEW GENERATION WITH 3D INFORMATION

The overview of MAGICDRIVE is depicted in Figure 3. Operating on the LDM pipeline, MAGICDRIVE generates street-view images conditioned on both scene annotations (**S**) and the camera pose (**P**) for each view. Given the 3D geometric information in scene annotations, projecting all to a BEV map, akin to BEVGen (Swerdlow et al., 2023) or BEVControl (Yang et al., 2023a), doesn't ensure precise guidance for street view generation, as exemplified in Figure 2. Consequently, MAGICDRIVE categorizes conditions into three levels: *scene* (text and camera pose), *foreground* (3D bounding boxes), and *background* (road map); and integrates them separately via cross-attention and an additive encoder branch, detailed in Section 4.1. Additionally, maintaining consistency across different cameras is crucial for synthesizing street views. Thus, we introduce a simple yet effective cross-view attention module in Section 4.2. Lastly, we elucidate our training strategies in Section 4.3, emphasizing Classifier-Free Guidance (CFG) in integrating various conditions.

### 4.1 GEOMETRIC CONDITIONS ENCODING

As illustrated in Figure 3, two strategies are employed for information injection into the UNet of diffusion models: cross-attention and additive encoder branch. Given that the attention mechanism (Vaswani et al., 2017) is tailored for sequential data, cross-attention is apt for managing variable length inputs like text tokens and bounding boxes. Conversely, for grid-like data, such as road maps, the additive encoder branch is effective in information injection (Zhang et al., 2023a). Therefore, MAGICDRIVE employs distinct encoding modules for various conditions.

**Scene-level Encoding** includes camera pose $\mathbf{P} = \{\mathbf{K} \in \mathbb{R}^{3\times3}, \mathbf{R} \in \mathbb{R}^{3\times3}, \mathbf{T} \in \mathbb{R}^{3\times1}\}$, and text sequence **L**. For text, we construct the prompt with a template as "A driving scene image at {location}. {description} ", and leverage a pre-trained CLIP text encoder ($E_{text}$) as LDM (Rombach et al., 2022), as shown by Equation 2, where $L$ is the token length of **L**. As for camera pose, we first concat each parameter by their column, resulting in $\bar{\mathbf{P}} = [\mathbf{K}, \mathbf{R}, \mathbf{T}]^T \in \mathbb{R}^{7\times3}$. Since $\bar{\mathbf{P}}$ contains values from sin/cos functions and also 3D offsets, to have the model effectively interpret these high-frequency variations, we apply Fourier embedding (Mildenhall et al., 2020) to each 3-dim vector before leveraging a Multi-Layer Perception (MLP, $E_{cam}$) to embed the camera pose parameters, as in Equation 3. To maintain consistency, we set the dimension of $h^c$ the same as that of $h_i^t$. Through the CLIP text encoder, each text embedding $h_i^t$ already contains positional information (Radford et al., 2021). Therefore, we prepend the camera pose embedding $h^c$ to text embeddings, resulting in scene-level embedding $\boldsymbol{h}^s = [h^c, \boldsymbol{h}^t]$.

$$\boldsymbol{h}^t = [h_1^t \dots h_L^t] = E_{text}(\mathbf{L}), \tag{2}$$

$$h^c = E_{cam}(\text{Fourier}(\bar{\mathbf{P}})) = E_{cam}(\text{Fourier}([\mathbf{K}, \mathbf{R}, \mathbf{T}]^T)). \tag{3}$$

**3D Bounding Box Encoding.** Since each driving scene has a variable length of bounding boxes, we inject them through the cross-attention mechanism similar to scene-level information. Specifically, we encode each box into a hidden vector $h^b$, which has the same dimensions as that of $h^t$. Each 3D bounding box $(c_i, b_i)$ contains two types of information: class label $c_i$ and box position $b_i$. For

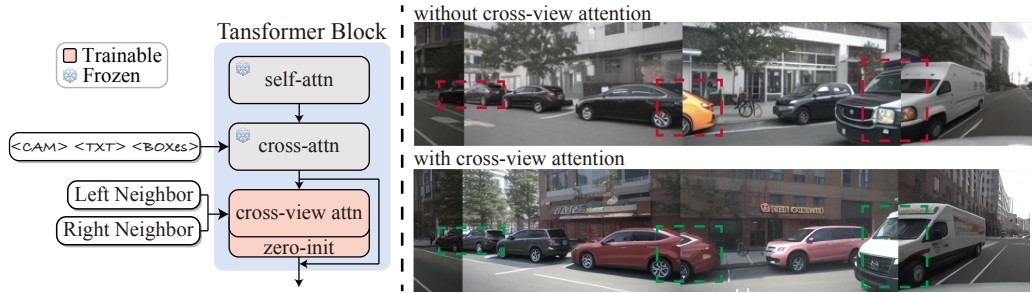

Figure 4: Cross-view Attention. *left*: we introduce cross-view attention to the pre-trained UNet after the cross-attention module. *right*: we highlight some areas for comparison between without and with cross-view attention. Cross-view attention guarantees consistency across multiple views.

class labels, we utilize the method similar to Li et al. (2023b), where the pooled embeddings of class names ($L_{c_i}$) are considered as label embeddings. For box positions $b_i \in \mathbb{R}^{8 \times 3}$, represented by the coordinates of its 8 corner points, we utilize Fourier embedding to each point and pass through an MLP for encoding, as in Equation 4. Then, we use an MLP to compress both class and position embedding into one hidden vector, as in Equation 5. The final hidden states for all bounding boxes of each scene are represented as $\boldsymbol{h}^b = [h_1^b \dots h_N^b]$, where $N$ is the number of boxes.

$$e_c^b(i) = \mathrm{AvgPool}(E_{text}(L_{c_i})), \ e_p^b(i) = \mathrm{MLP}_p(\mathrm{Fourier}(b_i)), \tag{4}$$

$$h_i^b = E_{box}(c_i, b_i) = \mathrm{MLP}_b(e_c^b(i), e_p^b(i)). \tag{5}$$

Ideally, the model learns the geometric relationship between bounding boxes and camera pose through training. However, the distribution of the number of visible boxes to different views is long-tailed. Thus, we bootstrap learning by filtering visible objects to each view ($v_i$), *i.e.*, $f_{viz}$ in Equation 6. Besides, we also add invisible boxes for augmentation (more details in Section 4.3).

$$\boldsymbol{h}_{v_i}^b = \{h_i^b \in \boldsymbol{h}^b | f_{viz}(b_i, \mathbf{R}_{v_i}, \mathbf{T}_{v_i}) > 0\}. \tag{6}$$

**Road Map Encoding.** The road map has a 2D-grid format. While Zhang et al. (2023a) shows the addictive encoder can incorporate this kind of data for 2D guidance, the inherent perspective differences between the road map's BEV and the camera's First-Person View (FPV) create discrepancies. BEVControl (Yang et al., 2023a) employs a back-projection to transform from BEV to FPV but complicates the situation with an ill-posed problem. In MAGICDRIVE, we propose that explicit view transformation is unnecessary, as sufficient 3D cues (*e.g.*, height from object boxes and camera pose) allow the addictive encoder to accomplish view transformation. Specifically, we integrate scene-level and 3D bounding box embeddings into the map encoder (see Figure 3). Scene-level embeddings provide camera poses, and box embeddings offer road elevation cues. Additionally, incorporating text descriptions facilitates the generation of roads under varying conditions (*e.g.*, weather and time of day). Thus, the map encoder can synergize with other conditions for generation.

## 4.2 CROSS-VIEW ATTENTION MODULE

In multi-camera view generation, it is crucial that image synthesis remains consistent across different perspectives. To maintain consistency, we introduce a cross-view attention module (Figure 4). Given the sparse arrangement of cameras in driving contexts, each cross-view attention allows the target view to access information from its immediate left and right views, as in Equation 7; here, $t$, $l$, and $r$ are the target, left, and right view respectively. Then, the target view aggregates such information with skip connection, as in Equation 8, where $\boldsymbol{h}^v$ indicates the hidden state of the target view.

$$\mathrm{Attention}_{cv}^i(Q_t, K_i, V_i) = \mathrm{softmax}(\tfrac{Q_t K_i^T}{\sqrt{d}}) \cdot V_i, \ i \in \{l, r\}, \tag{7}$$

$$\boldsymbol{h}_{out}^v = \boldsymbol{h}_{in}^v + \mathrm{Attention}_{cv}^l + \mathrm{Attention}_{cv}^r. \tag{8}$$

We inject cross-view attention after the cross-attention module in the UNet and apply zero-initialization (Zhang et al., 2023a) to bootstrap the optimization. The efficacy of the cross-view attention module is demonstrated in Figure 4 right, Figure 5, and Figure 6. The multilayered structure of UNet enables aggregating information from long-range views after several stacked blocks. Therefore, using cross-view attention on adjacent views is enough for multi-view consistency, further evidenced by the ablation study in Appendix C.

### 4.3 MODEL TRAINING

**Classifier-free Guidance** reinforces the impact of conditional guidance (Ho & Salimans, 2021; Rombach et al., 2022). For effective CFG, models need to discard conditions during training occasionally. Given the unique nature of each condition, applying a drop strategy is complex for multiple conditions. Therefore, our MAGICDRIVE simplifies this for four conditions by concurrently dropping scene-level conditions (camera pose and text embeddings) at a rate of $\gamma^s$. For boxes and maps, which have semantic representations for *null* (i.e., padding token in boxes and 0 in maps) in their encoding, we maintain them throughout training. At inference, we utilize *null* for all conditions, enabling meaningful amplification to guide generation.

**Training Objective and Augmentation.** With all the conditions injected as inputs, we adapt the training objective described in Section 3 to the multi-condition scenario, as in Equation 9.

$$\ell = \mathbb{E}_{\boldsymbol{x}_0,\boldsymbol{\epsilon},t,\{\boldsymbol{S},\boldsymbol{P}\}} \left[ ||\boldsymbol{\epsilon} - \boldsymbol{\epsilon}_\theta(\sqrt{\bar{\alpha}_t}\mathcal{E}(\boldsymbol{x}_0) + \sqrt{1-\bar{\alpha}_t}\boldsymbol{\epsilon}, t, \{\mathbf{S},\mathbf{P}\})|| \right]. \tag{9}$$

Besides, we emphasize two essential strategies when training our MAGICDRIVE. First, to counteract our filtering of visible boxes, we randomly add 10% invisible boxes as an augmentation, enhancing the model's geometric transformation capabilities. Second, to leverage cross-view attention, which facilitates information sharing across multiple views, we apply unique noises to different views in each training step, preventing trivial solutions to Equation 9 (*e.g.*, outputting the shared component across different views). Identical random noise is reserved exclusively for inference.

## 5 EXPERIMENTS

### 5.1 EXPERIMENTAL SETUPS

**Dataset and Baselines.** We employ the nuScenes dataset (Caesar et al., 2020), a prevalent dataset in BEV segmentation and detection for driving, as the testing ground for MAGICDRIVE. We adhere to the official configuration, utilizing 700 street-view scenes for training and 150 for validation. Our baselines are BEVGen (Swerdlow et al., 2023) and BEVControl (Yang et al., 2023a), both recent propositions for street view generation. Our method considers 10 object classes and 8 road classes, surpassing the baseline models in diversity. Appendix B holds additional details.

**Evaluation Metrics.** We evaluate both realism and controllability for street view generation. Realism is mainly measured using Fréchet Inception Distance (FID), reflecting image synthesis quality. For controllability, MAGICDRIVE is evaluated through two perception tasks: BEV segmentation and 3D object detection, with CVT (Zhou & Krähenbühl, 2022) and BEVFusion (Liu et al., 2023a) as perception models, respectively. Both of them are renowned for their performance in each task. Firstly, we generate images aligned with the validation set annotations and use perception models pre-trained with real data to assess image quality and control accuracy. Then, data is generated based on the training set to examine the support for training perception models as data augmentation.

**Model Setup.** Our MAGICDRIVE utilizes pre-trained weights from Stable Diffusion v1.5, training only newly added parameters. Per Zhang et al. (2023a), a trainable UNet encoder is created for $E_{map}$. New parameters, except for the zero-init module and the class token, are randomly initialized. We adopt two resolutions to reconcile discrepancies in perception tasks and baselines: 224×400 (0.25× down-sample) following BEVGen and for CVT model support, and a higher 272×736 (0.5× down-sample) for BEVFusion support. Unless stated otherwise, images are sampled using the UniPC (Zhao et al., 2023) scheduler for 20 steps with CFG at 2.0.

### 5.2 MAIN RESULTS

**Realism and Controllability Validation.** We assess MAGICDRIVE's capability to create realistic street-view images with the annotations from the nuScenes validation set. As shown by Table 1, MAGICDRIVE outperforms others in image quality, yielding notably lower FID scores. Regarding controllability, assessed via BEV segmentation tasks, MAGICDRIVE equals or exceeds baseline results at 224×400 resolution due to the distinct encoding design that enhances vehicle generation precision. At 272×736 resolution, our encoding strategy advancements enhance vehicle mIoU performance. Cropping large areas negatively impacts road mIoU on CVT. However, our bounding box encoding efficacy is backed by BEVFusion's results in 3D object detection.

Table 1: Comparison of generation fidelity with driving-view generation methods. Conditions for data synthesis are from nuScenes validation set. For each task, we test the corresponding models trained on the nuScenes training set. MAGICDRIVE surpasses all baselines throughout the evaluation. ↑/↓ indicates that a higher/lower value is better. The best results are in **bold**, while the second best results are in *underlined italic* (when other methods are available).

| Method | Synthesis resolution | FID↓ | BEV segmentation | | 3D object detection | |
|---|---|---|---|---|---|---|
| | | | Road mIoU ↑ | Vehicle mIoU ↑ | mAP ↑ | NDS ↑ |
| Oracle | - | - | 72.21 | 33.66 | 35.54 | 41.21 |
| Oracle | 224×400 | - | 72.19 | 33.61 | 23.54 | 31.08 |
| BEVGen | 224×400 | 25.54 | 50.20 | 5.89 | - | - |
| BEVControl | - | 24.85 | *60.80* | 26.80 | - | - |
| MAGICDRIVE | 224×400 | **16.20** | **61.05** | *27.01* | 12.30 | 23.32 |
| MAGICDRIVE | 272×736 | *16.59* | 54.24 | **31.05** | **20.85** | **30.26** |

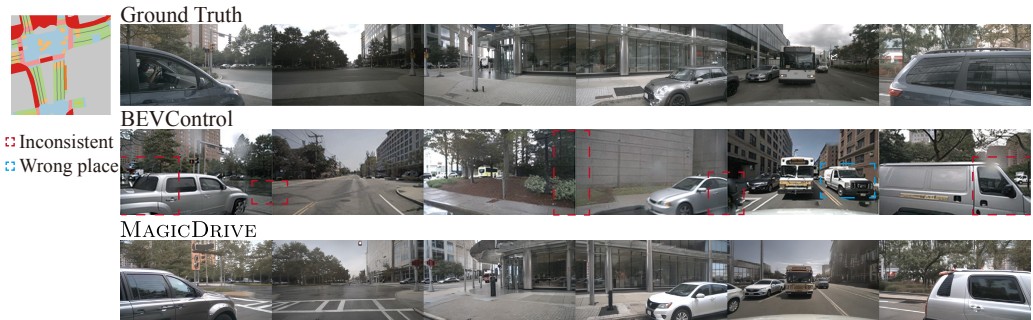

Figure 5: Qualitative comparison with BEVControl on driving scene from nuScenes validation set. We highlight some areas with rectangles to ease comparison. Compared with BEVControl, generations from MAGICDRIVE appear more consistent in both background and foreground.

**Training Support for BEV Segmentation and 3D Object Detection.** MAGICDRIVE can produce augmented data with accurate annotation controls, enhancing the training for perception tasks. For BEV segmentation, we augment an equal number of images as in the original dataset, ensuring consistent training iterations and batch sizes for fair comparisons to the baseline. As shown in Table 3, MAGICDRIVE significantly enhances CVT in both settings, outperforming BEVGen, which only marginally improves vehicle segmentation. For 3D object detection, we train BEVFusion models with MAGICDRIVE 's synthetic data as augmentation. To optimize data augmentation, we randomly exclude 50% of bounding boxes in each generated scene. Table 2 shows the advantageous impact of MAGICDRIVE 's data in both CAM-only (C) and CAM+LiDAR (C+L) settings. It's crucial to note that in CAM+LiDAR settings, BEVFusion utilizes both modalities for object detection, requiring more precise image generation due to LiDAR data incorporation. Nevertheless, MAGICDRIVE's synthetic data integrates seamlessly with LiDAR inputs, highlighting the data's high fidelity.

## 5.3 QUALITATIVE EVALUATION

**Comparison with Baselines.** We assessed MAGICDRIVE against two baselines, BEVGen and BEVControl, synthesizing multi-camera views for the same validation scenes (the comparison with BEVGen is in the Appendix D). Figure 5 illustrates that MAGICDRIVE generates images markedly superior in quality to BEVControl, particularly excelling in accurate object positioning and maintaining consistency in street views for backgrounds and objects. Such performance primarily stems from MAGICDRIVE 's bounding box encoder and its cross-view attention module.

**Multi-level Controls.** The design of MAGICDRIVE introduces multi-level controls to street-view generation through separation encoding. This section demonstrates the capabilities of MAGICDRIVE by exploring three control signal levels: *scene level* (time of day and weather), *background level* (BEV map alterations and conditional views), and *foreground level* (object orientation and deletion). As illustrated in Figure 1, Figure 6, and Appendix E, MAGICDRIVE adeptly accommodates alterations at each level, maintaining multi-camera consistency and high realism in generation.

Table 2: Comparison about support for 3D object detection model (*i.e.*, BEVFusion). MAGICDRIVE generates 272×736 images for augmentation. Results are reported on the nuScenes validation set.

| Modality | Data | mAP ↑ | NDS ↑ |
|---|---|---|---|
| C | w/o synthetic data | 32.88 | 37.81 |
| | w/ MAGICDRIVE | **35.40** +2.52 | **39.76** +1.95 |
| C+L | w/o synthetic data | 65.40 | 69.59 |
| | w/ MAGICDRIVE | **67.86** +2.46 | **70.72** +1.13 |

Table 3: Comparison about support for BEV segmentation model (*i.e.*, CVT). Results are reported by testing on the nuScenes validation set.

| Data | Vehicle mIoU ↑ | Road mIoU ↑ |
|---|---|---|
| w/o synthetic data | 36.00 | 74.30 |
| w/ BEVGen | 36.60 +0.60 | 71.90 -2.40 |
| w/ MAGICDRIVE | **40.34** +4.34 | **79.56** +5.26 |

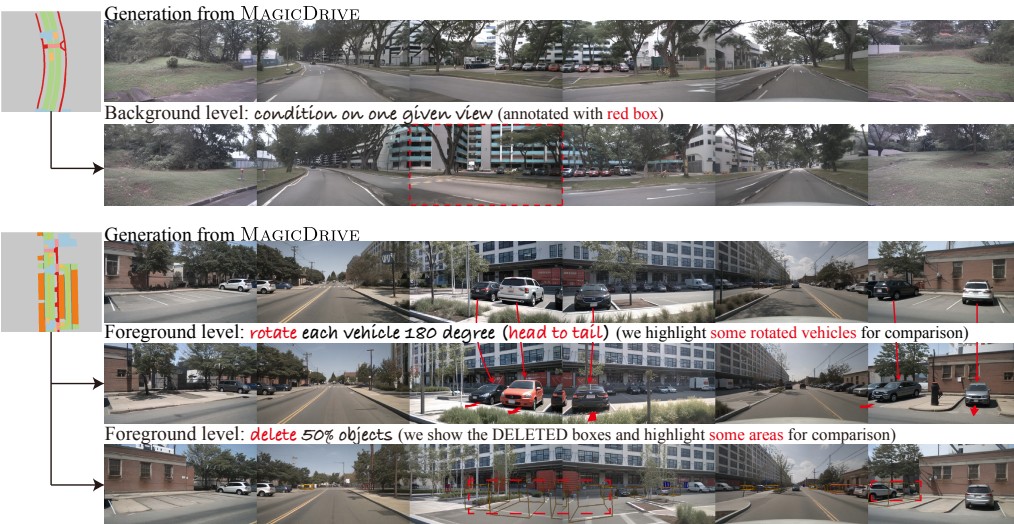

Figure 6: Showcase for multi-level control with MAGICDRIVE. We show background-level and foreground-level control separately with different conditions. All scenes are based on the nuScenes validation set. More results are in Appendix E.

## 5.4 EXTENSION TO VIDEO GENERATION

We demonstrate the extensibility of MAGICDRIVE to video generation by fine-tuning it on nuScenes videos. This involves modifying self-attention to ST-Attn (Wu et al., 2023a), adding a temporal attention module to each transformer block (Figure 7 *left*), and tuning the model on 7-frame clips with only the first and the last frames having bounding boxes. We sample initial noise independently for each frame using the UniPC (Zhao et al., 2023) sampler for 20 steps and illustrate an example in Figure 7 *right*.

Furthermore, by utilizing the interpolated annotations from ASAP (Wang et al., 2023c) like Drive-Dreamer (Wang et al., 2023b), MagicDrive can be extended to 16-frame video generation at 12Hz trained on Nvidia V100 GPUs. More results (*e.g.*, video visualization) can be found on our website.

## 6 ABLATION STUDY

**Bounding Box Encoding.** MAGICDRIVE utilizes separate encoders for bounding boxes and road maps. To demonstrate the efficacy, we train a ControlNet (Zhang et al., 2023a) that takes the BEV map with both road and object semantics as a condition (like BEVGen), denoted as "w/o $E_{box}$" in Table 4. Objects in BEV maps are relatively small, which require separate $E_{box}$ for accurate vehicle annotations, as shown by the vehicle mIoU performance gap. Applying visible object filter $f_{viz}$ significantly improves both road and vehicle mIoU by reducing the optimization burden. A MAGICDRIVE variant incorporating $E_{box}$ with BEV of road and object semantics didn't enhance performance, emphasizing the importance of integrating diverse information through different strategies.

**Effect of Classifier-free Guidance.** We focus on the two most crucial conditions, *i.e.* object boxes and road maps, and analyze how CFG affects the performance of generation. We change CFG from 1.5 to 4.0 and plot the change of validation results from CVT in Figure 8. Firstly, by increasing

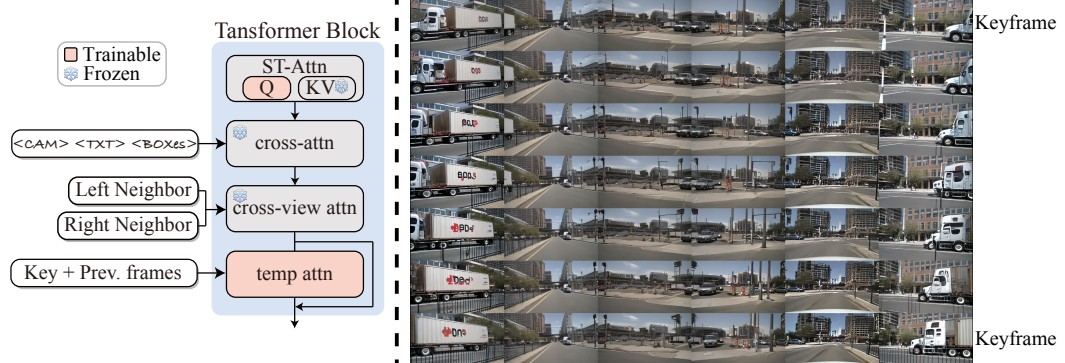

Figure 7: Extend MAGICDRIVE to video generation. *Left*: changes in transformer block for video generation. *Right*: result of video generation. Only keyframes have bounding box control.

Table 4: Ablation of the separate box encoder. Evaluation results are from CVT on the synthetic nuScenes validation set, without $M = \{0\}$ in CFG scale $= 2$. MAGICDRIVE has better controllability and keeps image quality.

| Method | FID $\downarrow$ | Road mIoU $\uparrow$ | Vehicle mIoU $\uparrow$ |
|---|---|---|---|
| w/o $E_{box}$ | 18.06 | 58.31 | 5.50 |
| w/o $f_{viz}$ | 14.67 | 56.46 | 24.73 |
| w/ $E_{box}$ & map$_{obj}$ | 14.70 | 56.04 | 26.20 |
| Ours | **14.46** | **59.31** | **27.13** |

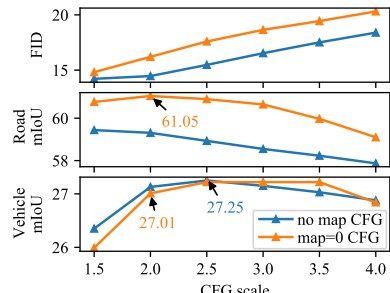

Figure 8: Effect of CFG on different conditions to each metrics.

CFG scale, FID degrades due to notable changes in contrast and sharpness, as seen in previous studies (Chen et al., 2023c). Secondly, retaining the same map for both conditional and unconditional inference eliminates CFG's effect on the map condition. As shown by blue lines of Figure 8, increasing CFG scale results in the highest vehicle mIoU at CFG=2.5, but the road mIoU keeps decreasing. Thirdly, with $M = \{0\}$ for unconditional inference in CFG, road mIoU significantly increases. However, it slightly degrades the guidance on vehicle generation. As mentioned in Section 4.3, CFG complexity increases with more conditions. Despite simplifying training, various CFG choices exist during inference. We leave the in-depth investigation for this case as future work.

## 7 CONCLUSION

This paper presents MAGICDRIVE, a novel framework to encode multiple geometric controls for high-quality multi-camera street view generation. With the separation encoding design, MAGICDRIVE fully utilizes geometric information from 3D annotations and realizes accurate semantic control for street views. Besides, the proposed cross-view attention module is simple yet effective in guaranteeing consistency across multi-camera views. As evidenced by experiments, the generations from MAGICDRIVE show high realism and fidelity to 3D annotations. Multiple controls equipped MAGICDRIVE with improved generalizability for the generation of novel street views. Meanwhile, MAGICDRIVE can be used for data augmentation, facilitating the training for perception models on both BEV segmentation and 3D object detection tasks.

**Limitation and Future Work.** We show failure cases from MAGICDRIVE in Figure 9. Although MAGICDRIVE can generate night views, they are not as dark as real images (as in Figure 9a). This may be due to that diffusion models are hard to generate too dark images (Guttenberg, 2023). Figure 9b shows that MAGICDRIVE cannot generate unseen weathers for nuScenes. Future work may focus on how to improve the cross-domain generalization ability of street view generation.

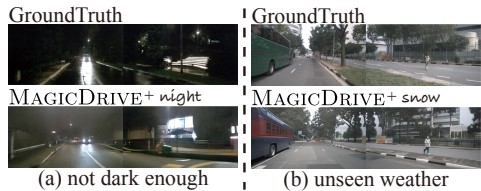

Figure 9: Failure cases of MAGICDRIVE.

**Acknowledgement.** This work is supported in part by the General Research Fund (GRF) of Hong Kong Research Grants Council (RGC) under Grant No. 14203521, in part by the CUHK SSFCRS funding No. 3136023, and in part by the Research Matching Grant Scheme under Grant No. 7106937, 8601130, and 8601440. We gratefully acknowledge the support of MindSpore, CANN (Compute Architecture for Neural Networks) and Ascend AI Processor used for this research. This research has been made possible by funding support from the Research Grants Council of Hong Kong through the Research Impact Fund project R6003-21.

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

# APPENDIX

## A  OBJECT FILTERING

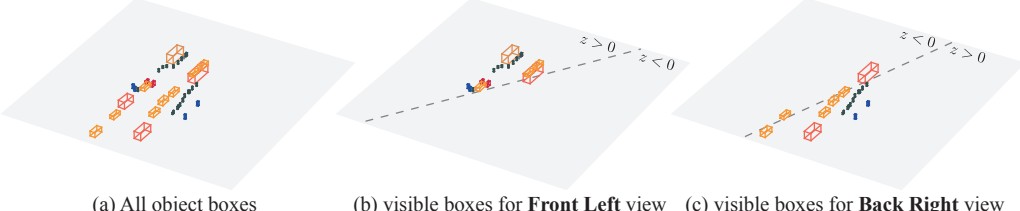

(a) All object boxes     (b) visible boxes for **Front Left** view     (c) visible boxes for **Back Right** view

Figure 10: Bounding box filtering to each view. The dashed lines in (b-c) represent the $x$-axis of the camera's coordinates. Boxes are retained only if they have at least a point in the positive half of the $z$-axis in each camera's coordinate.

In Equation 6, we employ $f_{viz}$ for object filtering to facilitate bootstrap learning. We show more details of $f_{viz}$ here. Refer to Figure 10 for illustration. For the sake of simplicity, each camera's Field Of View (FOV) is not considered. Objects are defined as *visible* if any corner of their bounding boxes is located in front of the camera (*i.e.*, $z^{v_i} > 0$) within each camera's coordinate system. The application of $f_{viz}$ significantly lightens the workload of the bounding box encoder, evidence for which can be found in Section 6.

## B  MORE EXPERIMENTAL DETAILS

**Semantic Classes for Generation.** To support most perception models on nuScenes, we try to include semantics commonly used in most settings (Huang et al., 2021; Zhou & Krähenbühl, 2022; Liu et al., 2023a; Ge et al., 2023). Specifically, for objects, ten categories include car, bus, truck, trailer, motorcycle, bicycle, construction vehicle, pedestrian, barrier, and traffic cone. For the road map, eight categories include drivable area, pedestrian crossing, walkway, stop line, car parking area, road divider, lane divider, and roadblock.

**Optimization.** We train all newly added parameters using AdamW (Loshchilov & Hutter, 2019) optimizer and a constant learning rate at $8e^{-5}$ and batch size 24 (total 144 images for 6 views) with a linear warm-up of 3000 iterations, and set $\gamma^s = 0.2$.

## C  ABLATION ON NUMBER OF ATTENDING VIEWS

Table 5: Ablation on number of attending views. Evaluation results are from CVT on the synthetic nuScenes validation set, without $M = \{0\}$ in CFG scale = 2.

| Attending Num. | FID $\downarrow$ | Road mIoU $\uparrow$ | Vehicle mIoU $\uparrow$ |
|---|---|---|---|
| 1 | **13.06** | 58.63 | 26.41 |
| 2 (ours) | 14.46 | **59.31** | **27.13** |
| 5 (all) | 14.76 | 58.35 | 25.41 |

In Table 5, we demonstrate the impact of varying the number of attended views on evaluation results. Attending to a single view yields superior FID results; the reduced influx of information from neighboring views simplifies optimization for that view. However, this approach compromises mIoU, also reflecting less consistent generation, as depicted in Figure 11. Conversely, incorporating all views deteriorates performance across all metrics, potentially due to excessive information causing interference in cross-attention. Since each view has an intersection with both left and right views, attending to one view cannot guarantee consistency, especially for foreground objects, while attending to more views requires more computation. Thus, we opt for 2 attended views in our main paper, striking a balance between consistency and computational efficiency.

Attend to 2 views    Attend to 1 view

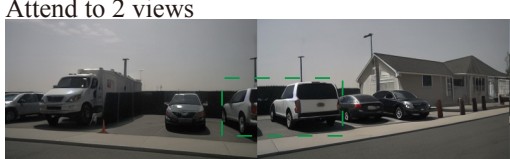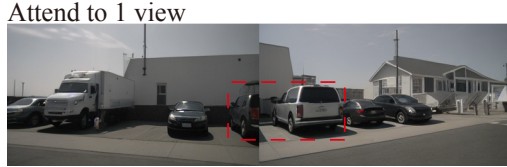

Figure 11: Comparison between different numbers of attending views. Only attending to one view results in worse multi-camera consistency.

## D    QUALITATIVE COMPARISON WITH BEVGEN

Figure 12 illustrates that MAGICDRIVE generates images with higher quality compared to BEV-Gen (Swerdlow et al., 2023), particularly excelling in objects. Such enhancement can be attributed to MAGICDRIVE's utilization of the diffusion model and the adoption of a customized condition injection strategy.

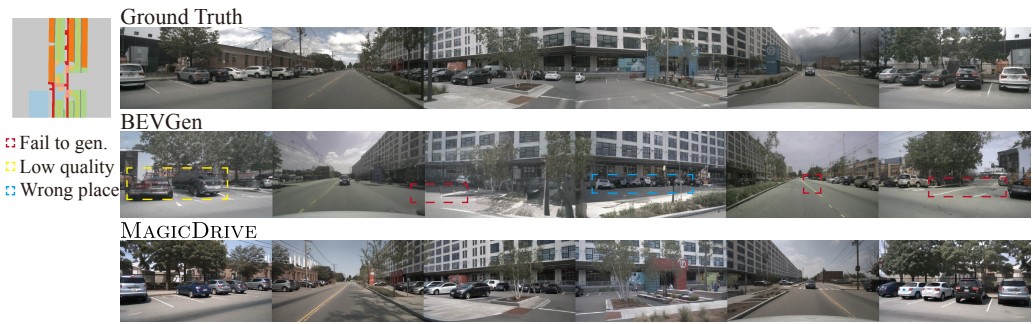

Figure 12: Qualitative comparison with BEVGen on driving scene from nuScenes validation set. We highlight some areas with rectangles to ease comparison. Compared with BEVGen, image quality of objects from MAGICDRIVE is much better.

## E    MORE RESULTS WITH CONTROL FROM DIFFERENT CONDITIONS

Figure 13 shows *scene level* control (time of day) and *background level* control (BEV map alterations). MAGICDRIVE can effectively reflect these changes in control conditions through the generated camera views.

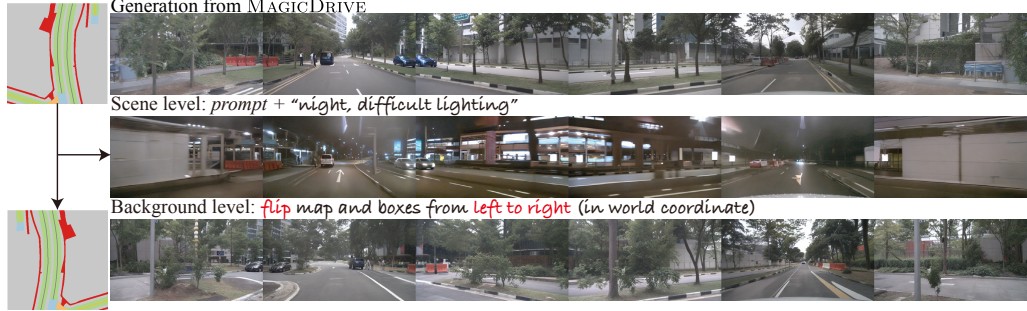

Figure 13: Showcase for scene-level control with MAGICDRIVE. The scene is from the nuScenes validation set.

## F    MORE EXPERIMENTS WITH 3D OBJECT DETECTION

Table 6: Comparison about support for 3D object detection model (*i.e.*, BEVFusion). MAGICDRIVE generates 272×736 images for augmentation. Results are from tests on the nuScenes validation set.

| Epoch | Data | CAM-Only | | CAM+LiDAR | |
|---|---|---|---|---|---|
| | | mAP ↑ | NDS ↑ | mAP ↑ | NDS ↑ |
| 0.5× | w/o synthetic data | 30.21 | 32.76 | too few epochs | too few epochs |
| | w/ MAGICDRIVE | 33.29 (+3.08) | 36.69 (+3.93) | | |
| 1× | w/o synthetic data | 32.88 | 37.81 | 65.40 | 69.59 |
| | w/ MAGICDRIVE | 35.40 (+2.52) | 39.76 (+1.95) | 67.86 (+2.46) | 70.72 (+1.13) |
| 2× | w/o synthetic data | 35.49 | 40.66 | 68.33 | 71.31 |
| | w/ MAGICDRIVE | 35.74 (+0.25) | 41.40 (+0.74) | 68.58 (+0.25) | 71.34 (+0.03) |

In Table 6, we show additional experimental results on training 3D object detection models using synthetic data produced by MAGICDRIVE. Given that BEVFusion utilizes a lightweight backbone (*i.e.*, Swin-T (Liu et al., 2021)), model performance appears to plateau with training through 1-2× epochs (2×: 20 for CAM-Only and 6 for CAM+LiDAR). Reducing epochs can mitigate this saturation, allowing more varied data to enhance the model's perceptual capacity in both settings. This improvement is evident even when epochs for 3D object detection are further reduced to 0.5×. Our MAGICDRIVE accurately augments street-view images with the annotations. Future works may focus on annotation sampling and construction strategies for synthetic data augmentation.

## G    MORE DISCUSSION

**More future work.**    Note that MAGICDRIVE-generated street views can currently only perform as augmented samples to train with real data, and it is exciting to train detectors solely with generated data, which will be explored in the future. More flexible usage of the generated street views beyond data augmentation, especially incorporation with generative pre-training (Chen et al., 2023a; Zhili et al., 2023), contrastive learning (Chen et al., 2021; Liu et al., 2022b) and the large language models (LLMs) (Chen et al., 2023b; Gou et al., 2023), is an appealing future research direction. It is also interesting to utilize the geometric controls in different circumstances beyond 3D scenarios (*e.g.*, multi-object tracking (Li et al., 2023a) and concept removal (Liu et al., 2023b)).

## H    DETAILED ANALYSIS ON 3D OBJECT DETECTION WITH SYNTHETIC DATA

Table 7: Per-class performance comparison with BEVFusion for 3D object detection with 1× setting. Results are tested on the nuScenes validation set.

| Data | mAP | car | cone | barrier | bus | ped. | motor. | truck | bicycle | trailer | constr. |
|---|---|---|---|---|---|---|---|---|---|---|---|
| BEVFusion | 32.88 | 50.67 | 50.46 | 48.62 | 37.73 | 35.74 | 30.40 | 27.54 | 24.85 | 15.56 | 7.28 |
| + MAGICDRIVE | 35.40 | 51.86 | 53.56 | 51.15 | 40.43 | 38.10 | 33.11 | 29.35 | 27.85 | 18.74 | 9.83 |
| Difference | +2.52 | +1.20 | +3.10 | +2.53 | +2.70 | +2.36 | +2.71 | +1.81 | +3.00 | +3.19 | +2.55 |

We provide per-class AP for 3D object detection from the nuScenes validation set using BEVFusion in Table 7. From the results, we observe that, firstly, the improvements for large objects are significant, for example, buses, trailers, and construction vehicles. Secondly, objects with less diverse appearances, such as traffic cones and barriers, show more improvement, especially compared to trucks. Thirdly, we note that the improvement is marginal for cars, while significant for pedestrians, motorcycles, and bicycles. This may be because the baseline already performs well for cars. For pedestrians, motorcycles, and bicycles, even though distant objects from the ego car are generated less faithfully, MAGICDRIVE can synthesize high-quality objects near the ego car, as shown in Figure 17-18. Therefore, more accurate detection of objects near the ego car contributes to improvements for these classes. Overall, mAP improvement comes with promotion in all classes' AP, indicating MAGICDRIVE can indeed help the training of perception models.

## I    MORE RESULTS FOR BEV SEGMENTATION

BEVFusion is also capable of BEV segmentation and considers most of the classes we used in the BEV map condition. Due to the lack of baselines, we present the results in Table 8 to facilitate comparison for future works. As can be seen, the 272×736 resolution does not outperform the 224×400 resolution. This is consistent with the results from CVT in Table 1 on the Road segment. Such results confirm that better map controls rely on maintaining the original aspect ratio for generation training (*i.e.*, avoiding cropping on each side).

Table 8: Generation fidelity to BEV map conditions. Results are tested with BEVFusion for BEV segmentation on the nuScenes validation set.

| Methods | resolution | mIOU for 6 classes | |
| --- | --- | --- | --- |
| | | CAM-only | CAM+LiDAR |
| Oracle | - | 57.09 | 62.94 |
| Oracle | 224×400 | 52.72 | 58.49 |
| MAGICDRIVE | 224×400 | **30.24** | **48.21** |
| MAGICDRIVE | 272×736 | 28.71 | 47.12 |

## J    GENERALIZATION OF CAMERA PARAMETERS

To improve generalization ability, MAGICDRIVE encodes raw camera intrinsic and extrinsic parameters for different perspectives. However, the generalization ability is somewhat limited due to nuScenes fixing camera poses for different scenes. Nevertheless, we attempt to exchange the intrinsic and extrinsic parameters between the three front cameras and three back cameras. The comparison is shown in Figure 14. Since the positions of the nuScenes cameras are not symmetrical from front to back, and the back camera has a 120° FOV compared to the 70° FOV of the other cameras, clear differences between front and back views can be observed for the same 3D coordinates.

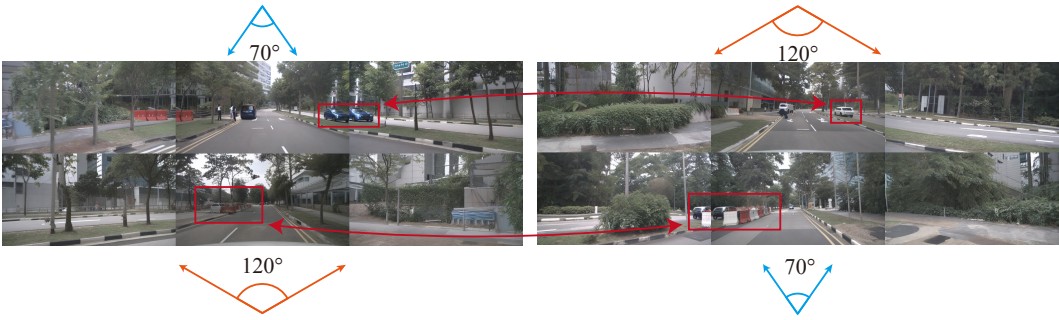

Figure 14: To show the generalization ability of learned camera encoding, we exchange the camera parameters between 3 front cameras and 3 back cameras. The 3D position is the same for two generations. We highlight some areas (with box boxes)

## K  MORE GENERATION RESULTS

We show some corner-case (Li et al., 2022) generations in Figure 15, and more generations in Figure 16-Figure 18.

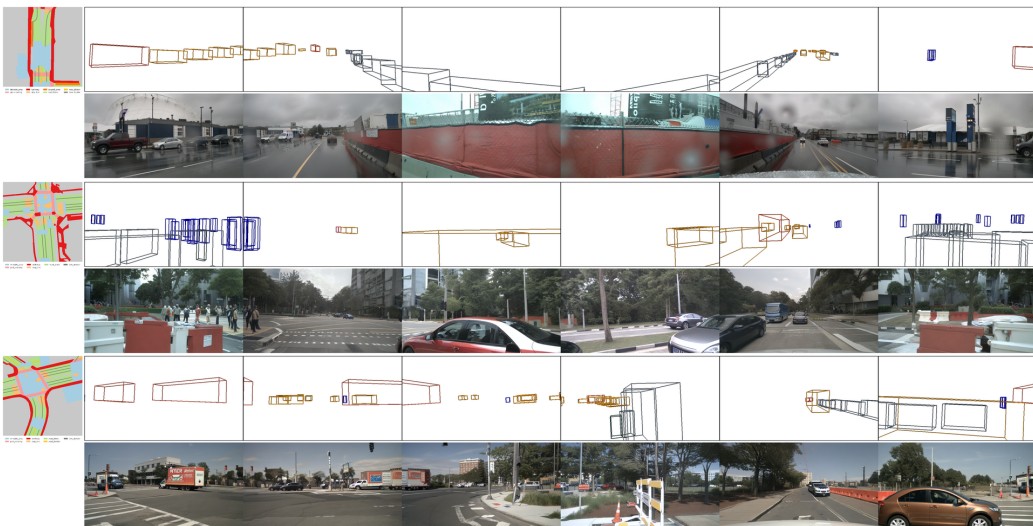

Figure 15: Generation from MAGICDRIVE with corner-case annotations.

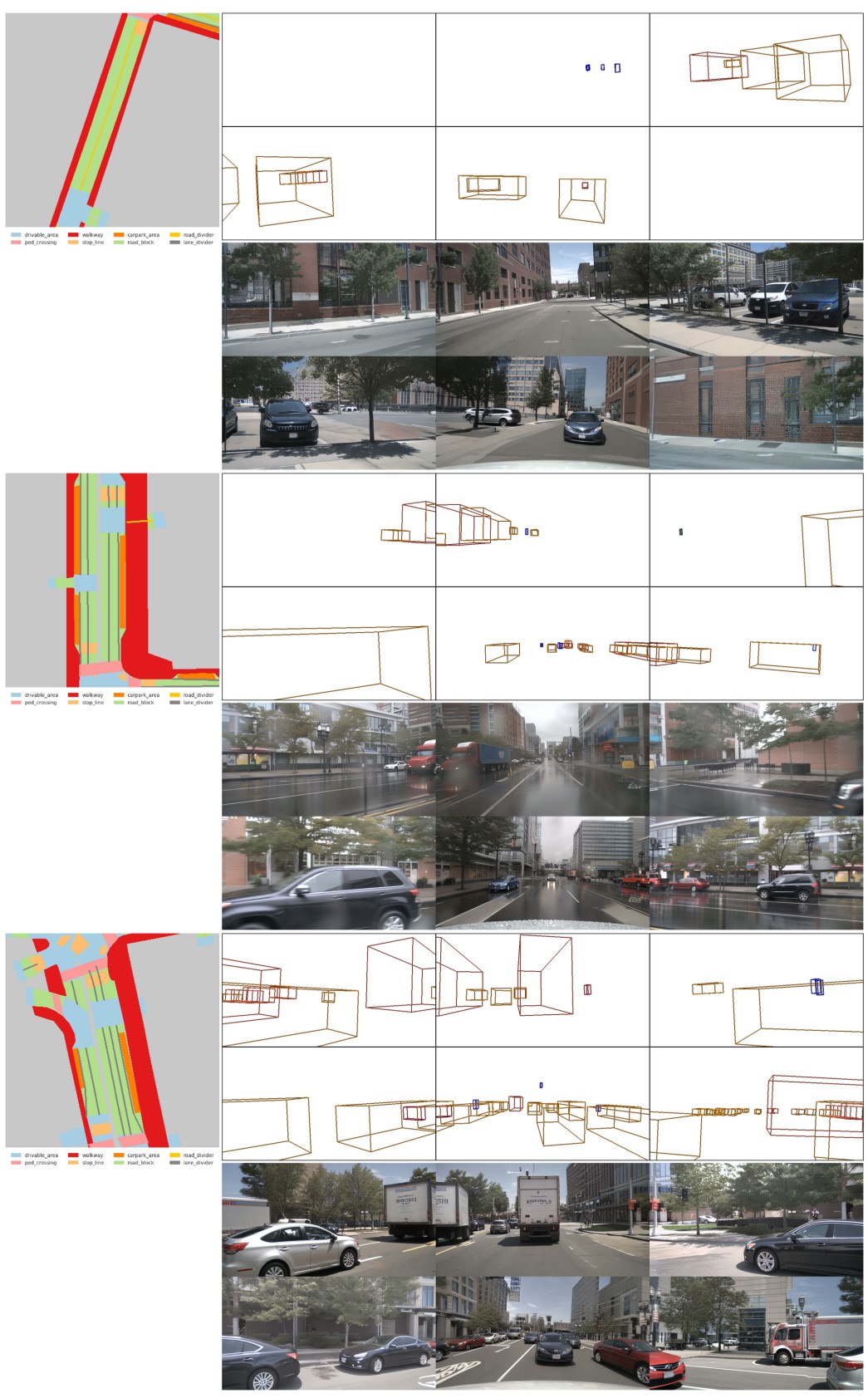

Figure 16: Generation from MAGICDRIVE with annotations from nuScenes validation set.

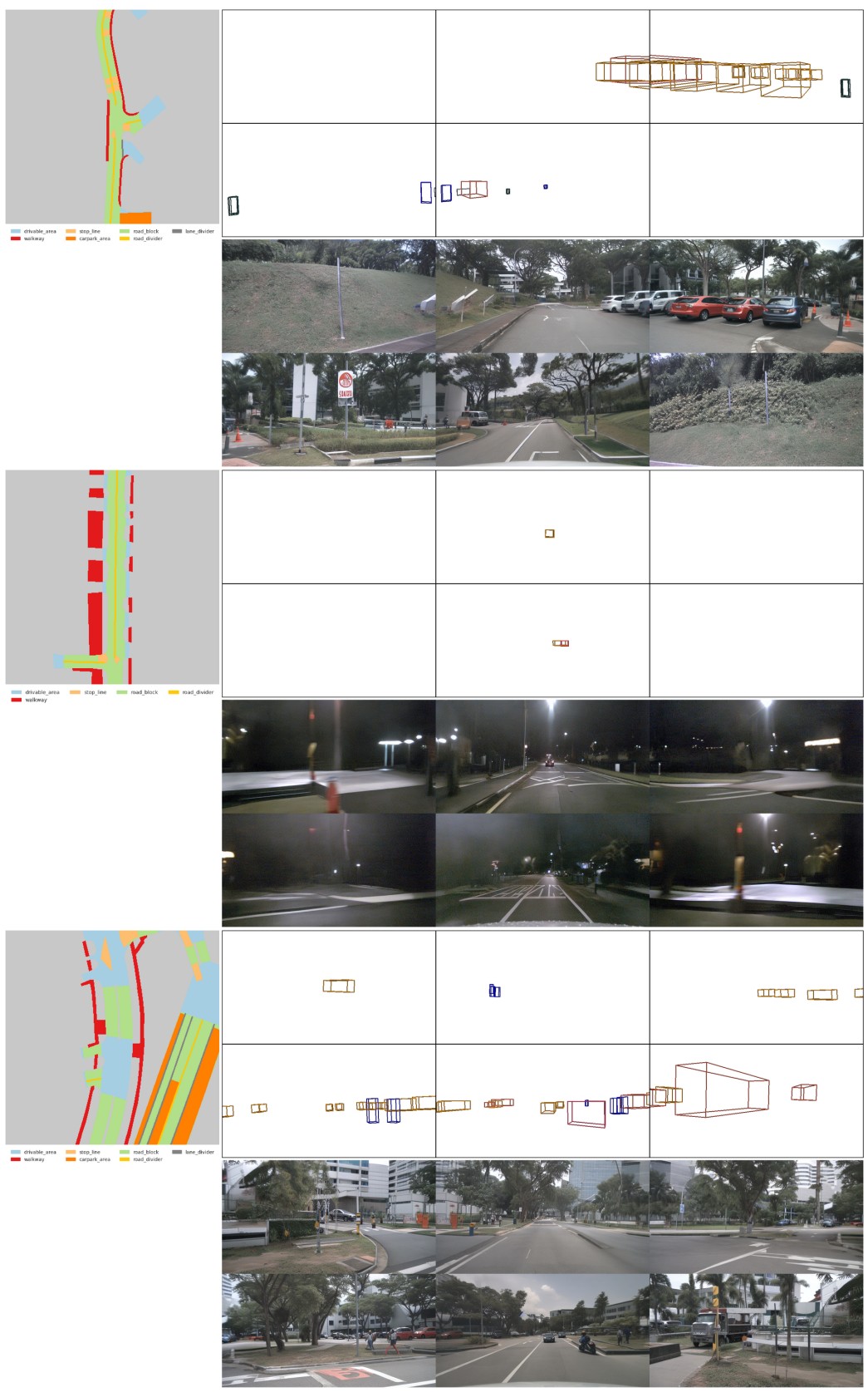

Figure 17: Generation from MAGICDRIVE with annotations from nuScenes validation set.

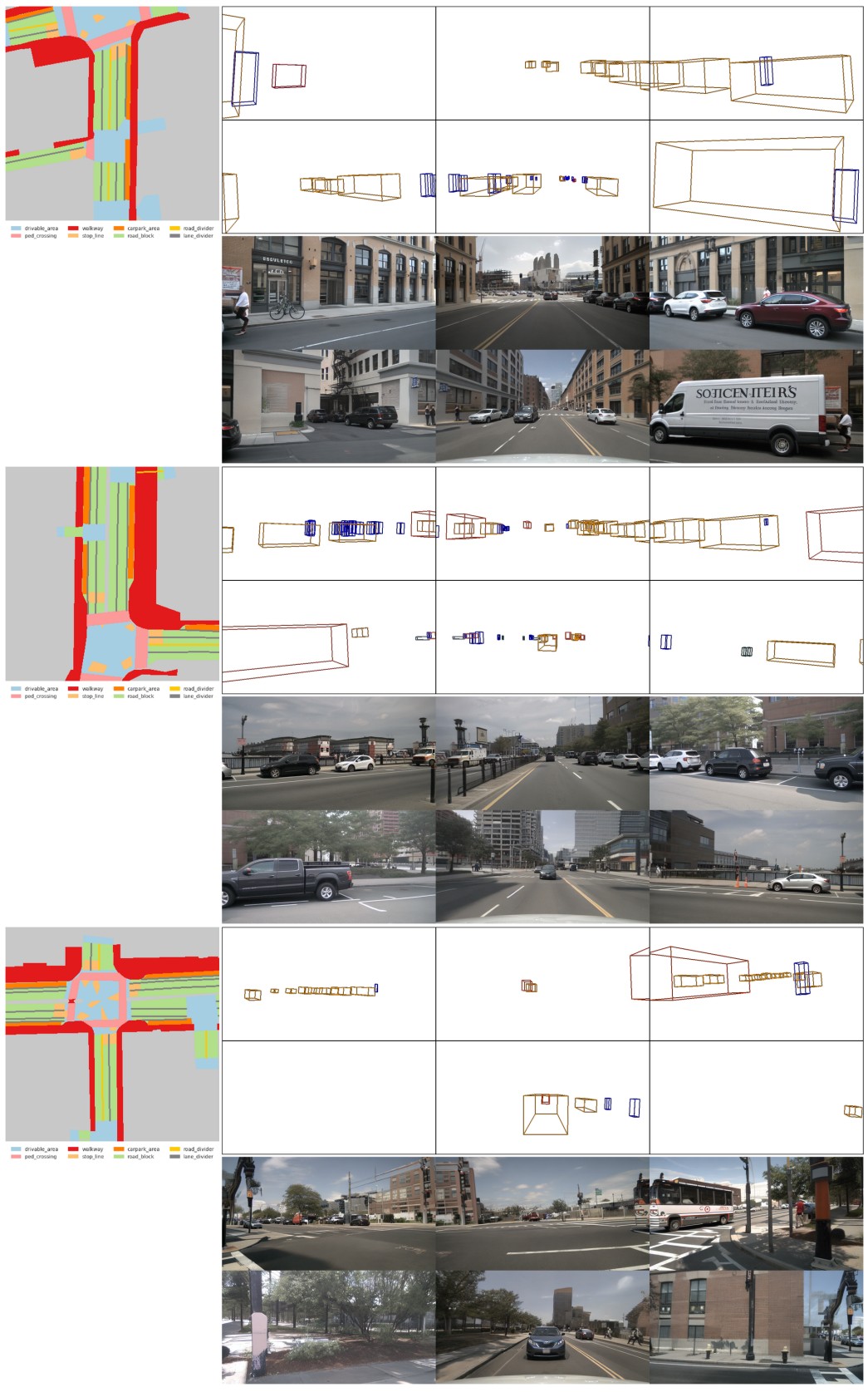

Figure 18: Generation from MAGICDRIVE with annotations from nuScenes validation set.

