# OpenReview forum: "MagicDrive: Street View Generation with Diverse 3D Geometry Control"
_ICLR.cc/2024/Conference — ICLR 2024 poster_

### Official Review · Reviewer_TY6p · 2023-10-31

**Soundness:** 3 good
**Presentation:** 3 good
**Contribution:** 2 fair
**Rating:** 6
**Confidence:** 2

**Summary:**

This work focus on street view generation conditioned on BEV layout.
The authors harness the power of pretrained stable diffusion model and ControlNet to generate realistic images.
Cross view attention is applied to achieve better multi-view consistency.
Experiments are conducted on nuscenes to demonstrate superior realism over the baselines.
The generative results can serve as data augmentation for downstream task to boost the performance.

**Strengths:**

1. Decent and realistic results. Some images are hard to distinguish unless zoom in.
2. The overall pipeline is sound to me, and compare to the baselines, it shows improved realism and better multi-camera consistency.
3. It can boost downstream perception performance.
4. Clear writing, easy to understand

**Weaknesses:**

1. The paper’s claim of “geometry control” appears to be somewhat overstated. Geometry encompasses more than just pose and Box3D; it also includes shape, topology, and the ability to modify any object within the scene.
2. The consistency aspect of the results is not fully realized. While I acknowledge that the multi-camera consistency is superior to that of the baselines, the broader aspect of consistency, such as consistency from novel viewpoints (e.g., moving cameras away from the original view, rotating it 360 degrees, modifying the focal length to zoom into details in distant regions), seems to be lacking. Based on my observations from the website and deductions from the approach, achieving such consistency with the current representation seems highly unlikely.
3. The novelty of this work is unclear to me, as I am not very familiar with this topic. Upon a quick review of BEVGen and BEVControl, it appears that the main difference lies in the new modeling of 3D bounding boxes (in this work, the authors decouple the 3D boxes and road maps and model them separately), the use of Stable Diffusion, and cross-view attention. However, none of these elements seem to be significantly innovative.

**Questions:**

Equation 9 is not closed in $||$

What if apply the data augmentation to the SOTA camera 3D detection models, can you achieve the new SOTA on nuscenes?

---

> ### Author Response · Authors · 2023-11-16
> **Response to Reviewer TY6p**
>
> Dear Reviewer TY6p,
>
> We greatly appreciate your time and insightful feedback. Below is our response to the identified weaknesses and questions.
>
> ---
>
> ### Weakness 1: the claim of “geometry control” appears to be somewhat overstated.
>
> Thank you for highlighting this concern. To clarify, we have added a footnote on page 2 that specifically defines the scope of "geometry control" as considered in this study. We hope this addition provides a clearer understanding of the term.
>
> ---
>
> ### Weakness 2: the consistency aspect of the results is not fully realized.
>
> MagicDrive, like the MLP in NeRF, uses raw camera pose as an input, enabling potential generalization with a diverse range of camera poses. For instance, by swapping back and front cameras, we have observed generalizations across different FOVs and extrinsic parameters (detailed in Appendix G). However, due to the fixed camera configuration in the nuScenes dataset, which limits variation, our model currently does not generalize to any camera pose. With a more diverse dataset of camera poses, we anticipate enhanced generalization capabilities.
>
> ---
>
> ### Weakness 3: novelty about the paper compared with BEVGen and BEVControl.
>
> Street view generation presents unique challenges, distinct from other text/pixel-level conditional generations. This complexity arises not only from the increased number of conditions but also from the unique format and perspective requirements, including the crucial aspect of multi-view consistency.
>
> 1. **Difference in handling conditions**: Compared to BEVGen, MagicDrive employs a distinct framework and produces significantly higher-quality data. BEVControl, on the other hand, projects 3D annotations to image view for ControlNet guidance, which loses crucial occlusion relationships and multi-view connections. In contrast, MagicDrive leverages raw 3D bounding boxes and BEV maps for encoding, leading to enhanced controllability, as demonstrated in Figure 5. Our innovative encoding method maintains its novelty and effectiveness even when compared to later developments like DrivingDiffusion [1].
> 2. **Enhanced Multi-View Consistency**: Street views have a limited number of cameras and different camera configurations (e.g., exposure). Therefore, the novelty lies in proposing a simple module that effectively solves the problem. Despite BEVGen and BEVControl employing more complex methods, their results are relatively modest compared to MagicDrive (as shown in Figure 5).
>
> Given that BEVControl was released in August, our work should be viewed as concurrent. Additionally, the effectiveness of our proposed methods is further substantiated by the ablations in Section 6 and Appendix C.
>
> ---
>
> ### Question 1: typos.
>
> We thank the reviewer for pointing out the issue. We have fixed the typo.
>
> ---
>
> ### Question 2: data augmentation for SOTA camera 3D detection models.
>
> Our paper primarily focuses on the controllable generation of high-quality street views. The training on downstream tasks is presented as a proof of concept, demonstrating the potential of synthetic data to enhance 3D perception models. For this purpose, we selected a strong and influential baseline model, BEVFusion, which remains a formidable contender in camera 3D detection when considering single-frame input.
>
> We acknowledge that achieving new SOTA nuScenes often involves extensive engineering efforts, such as utilizing unannotated data [2-3] and applying test-time augmentation. Therefore, our approach is to compare with a reliable baseline, rather than striving for new SOTA results. This comparison is intended to underscore the effectiveness of our method in a realistic and pragmatic context.
>
> ---
>
> [1] X. Li, Y. Zhang and X. Ye. DrivingDiffusion: Layout-Guided multi-view driving scene video generation with latent diffusion model. arXiv preprint arXiv:2310.07771, 2023.
>
> [2] X. Lin, T. Lin, Z. Pei, L. Huang and Z. Su. Sparse4D v2: Recurrent Temporal Fusion with Sparse Model, 2023.
>
> [3] Z. Zong, D. Jiang, G. Song, Z. Xue, J. Su, H. Li and Y. Liu. Temporal Enhanced Training of Multi-view 3D Object Detector via Historical Object Prediction, 2023.

---

### Official Review · Reviewer_UU4E · 2023-10-31

**Soundness:** 3 good
**Presentation:** 4 excellent
**Contribution:** 3 good
**Rating:** 8
**Confidence:** 4

**Summary:**

This paper introduces a novel framework for generating street view imagery with diverse 3D geometry controls such as camera poses, road maps, and 3D bounding boxes, using tailored encoding strategies. Existing methods primarily focus on 2D control which limits their utility in 3D perception tasks essential for autonomous driving. This paper consider a BEV view input, and input these control through encoding each of the information and insert these in the cross attention inside the diffusion UNet. In order to ensure the consistency between different views. It also introduce the cross-view attention for the training.

**Strengths:**

- Adopting diffusion for driving view synthese which trying to solve the data limitation in corner cases for self-driving is important.

- The overall strategy is sound and the paper proposed reasonable ways to encoding different information. Including Scene-level Encoding,  3D Bounding Box Encoding,  Road Map Encoding, these encoding are well organized and normalized in inserting to the cross attention module.  It also enables the final multi-level control of the generation.

- Other modules such as cross-view module help in image synthesis consistency.

- The experiments, show that it outperforms the other baselines such as BEVGen and BEV-Control, for synthesizing multi-camera views.

**Weaknesses:**

- the synthesized views are impressive,  the experiments are conducted in 700 street-view scenes for training and 150 for validation, which is a much smaller scale than the real-world senario. Wonder how to possiblly make it generalizable for real world domain. Does this be helpful to improve the detection & other understanding tasks when the data is large.

- In addition, not only for the dark scene, many generated instances such as human can be distorted with diffusion models. Wonder how that affects the detection accuracy for each subclass. The author provides overall accuracy in 3D object detection, may also analysis the details how to mix the synthesized images and real-images for training the model.

**Questions:**

Diffusion models are costly, Could the framework be extended or modified to handle real-time or near real-time massive generation requirements, which are crucial for applications in autonomous driving systems? This also related to handle dynamic entities in the scene such as moving vehicles or pedestrians, especially when synthesizing views over a period of time?

How closely does the synthetic data correlate with real-world data and what measures were taken to ensure the accuracy and reliability of this synthetic data?

---

> ### Author Response · Authors · 2023-11-16
> **Response to Reviewer UU4E (1/2)**
>
> Dear Reviewer UU4E,
>
> We appreciate your time and valuable insights. Below is our response to the weaknesses and questions.
>
> ---
> ### Weakness 1: limited scale of the dataset used in experiments and the generalizability of MagicDive to real-world scenarios.
> NuScenes, a well-established real-world dataset for autonomous driving, forms the basis of our training, allowing MagicDrive to demonstrate strong generalization across various scenarios, including unseen objects' positions and maps. This adaptability is evident from our tests involving manual modifications of object boxes (rotation, deletion, and moving), as detailed in Figure 6. While the current dataset scale is smaller than real-world scenarios, our results already show promising generalization capabilities. Scaling up the dataset size would likely present a broader spectrum of real-world scenarios, enhancing the model's performance even further. It is expected that such an expansion would significantly enhance MagicDrive's utility in downstream perception tasks.
>
> ---
> ### Weakness 2: effects of distortion in generated images on the detection accuracy for each subclass and details on how synthetic images are integrated with real images.
>
> Thank you for your insightful comments. In response, we have expanded Appendix E to include a subclass-specific analysis of detection accuracy. This analysis demonstrates that the integration of synthetic data, even with instances of distortion, positively impacts the detection performance of each subclass. For instance, pedestrian detection shows a notable improvement of +2.36 mAP. This suggests that despite some distortions, the high-quality rendering of objects, particularly those close to the ego car, substantially aids in enhancing perception model accuracy.
>
> In our experiments, we generate an equal number of samples as the training set for data augmentation in BEV segmentation. For 3D object detection, we randomly delete 50% of objects in each scene for generation, as detailed in Sec. 5.2. Exploring various strategies for integrating real and synthetic data is indeed a worthy endeavor. While our current focus is on advancing high-quality street view generation, we acknowledge the importance of this area and leave its detailed exploration to future research.
>
> ---
> ### Question 1: feasibility of adapting MagicDrive for real-time generation and how it manages dynamic entities over time.
>
> 1. **Regarding Real-time Generation.** Our current model is optimized more for offline applications, such as simulation and data augmentation, with a focus on realism and controllability rather than low latency. However, we are aware of the importance of real-time processing in autonomous driving. The generation time in our current setup is approximately 3 seconds for 6 views using a single V100 card. Although this is not yet real-time, advancements in inference acceleration (e.g., advanced sampling techniques, model distillation methods like the Latent Consistency Model [1], or engineering solutions like low-bit inference [2] and flash attention [3]) are promising areas for reducing this latency. These potential improvements are areas we plan to explore in future research to make our model more applicable to real-time scenarios.
> 2. **Handling Dynamic Entities.** To address the dynamic nature of autonomous driving scenes, we are exploring the integration of video generation techniques, such as tune-a-video [4], into MagicDrive. This would allow the model to account for moving entities over time. Preliminary results from these explorations show promise, and we have included a demonstration video in Appendix F and on our [anonymous website](https://magic-drive.github.io/). This indicates that adapting MagicDrive to dynamically changing environments is feasible.

---

> > ### Author Response · Authors · 2023-11-16
> > **Response to Reviewer UU4E (2/2)**
> >
> > ### Question 2: measurement of generation accuracy and reliability
> >
> > We employ two evaluation approaches, as detailed in Table 1. Firstly, traditional generation metrics (such as FID) assess the congruence between real and synthetic data distributions. Secondly, we utilize pre-trained perception models (BEVFusion for 3D object detection and CVT for BEV segmentation), originally trained on real data. The closer our results align with real data, the better. Additionally, the "C+L" improvement noted in Table 2 indicates a high degree of consistency between generated objects and LiDAR points, further evidencing our model's superior controllability.
> >
> > ------
> >
> > [1] S. Luo, Y. Tan, L. Huang, J. Li, and H. Zhao. Latent consistency models: Synthesizing high-resolution images with few-step inference, 2023.
> >
> > [2] Y. He, L. Liu, J. Liu, W. Wu, H. Zhou, B. Zhuang. PTQD: Accurate Post-Training Quantization for Diffusion Models, 2023.
> >
> > [3] T. Dao. FlashAttention-2: Faster Attention with Better Parallelism and Work Partitioning, 2023.
> >
> > [4] J. Z. Wu, Y. Ge, X. Wang, S. W. Lei, Y. Gu, Y. Shi, W. Hsu, Y. Shan, X. Qie, and M. Z. Shou. Tune-a-video: One-shot tuning of image diffusion models for text-to-video generation. ICCV, 2023.

---

### Official Review · Reviewer_jXFZ · 2023-11-01

**Soundness:** 3 good
**Presentation:** 4 excellent
**Contribution:** 2 fair
**Rating:** 5
**Confidence:** 4

**Summary:**

The goal of this paper is to have multi-modality control on street scene generation process. Overall idea of this paper is to use ControlNet framework on top of pre-trained stable diffusion model to support conditional generation over street view dataset. The complexity comes in terms of how to design multi-modality conditions for the ControlNet condition. For this purpose, the authors introduce various cross attention over their conditions to fuse their conditions onto scene representation (they also feed non-scene related condition directly to stable diffusion). The training follows ControlNet paradigm with classifier-free guidance to encourage output more aligned with conditions. Result-wise, they compare with BEVGen and BEVControl on nuScenes. The experiment aims to reveal they produce more realistic images and have better control over output space for street view generation task.

**Strengths:**

1. The paper is very nicely organized and written.
2. The quality of the generated street view is realistic
3. We can see MagicDrive has more precise control on street generation than baselines

**Weaknesses:**

1. The main concern is for their marginal technical contribution. The proposed method is ControlNet applied into street view generation setting with multi-modality condition. The novelty probably lies in how to organize the condition into controlNet setting, which might not sufficient for acceptance.
2. MagicDrive does not ensure consistency across adjacent frames after checking their website demo.

**Questions:**

1. Do you have different CFG weights for different conditions? If so, I am curious on how you make that work.

---

> ### Author Response · Authors · 2023-11-16
> **Response to Reviewer jXFZ**
>
> Dear Reviewer jXFZ,
>
> We thank you for pointing out that "*The quality of the generated street view is realistic*" and "*We can see MagicDrive has more precise control on street generation than baselines*" as strengths. Here we provide our response to weaknesses and questions.
>
> ---
> ### Weaknesses 1: technical contributions compared to ControlNet
>
> Street view generation presents unique challenges, distinct from other conditional methods such as ControlNet. This complexity arises not only from the increased number of conditions but also from the unique format and perspective requirements, including the crucial aspect of multi-view consistency. Our technical contributions, detailed in Sec. 1, are summarized as follows:
>
> 1. **Utilization of Raw 3D Data in MagicDrive**: Unlike other works, MagicDrive is the first to employ raw BEV maps and objects' bounding boxes & camera pose with 3D coordinates for street view generation. This approach differs significantly from ControlNet for two main reasons:
>     - **Interactive Conditions**: The conditions are interdependent. For instance, the relevance of 3D bounding boxes is contingent on the camera pose, and the terrain height in BEV maps is inferred from the objects' positional heights.
>     - **Varying Coordinates/Perspectives**: We handle conditions presented in different coordinates and perspectives, such as BEV maps in BEV perspective and 3D objects and camera pose in LiDAR coordinates. All of them are distinct from the pixel-level conditions in ControlNet.
>
>     Actually, we only adapt ControlNet's method for map condition injection, and propose our own encoders for other conditions. Note that MagicDrive is the first to show that ControlNet is capable of encoding conditions from different perspectives. Besides, our tailored encoding of the box and camera, as evidenced in Sec. 6 and Table 4, is crucial for performance and multi-view consistency.
>
> 2. **Innovative Cross-View Attention Module**: To address the limited overlap and varying characteristics of each camera view (e.g., intrinsic differences, and exposure variation as shown in Figure 5), we developed a simple yet effective cross-view attention module. This module ensures consistency across views, a feature not addressed by ControlNet. As demonstrated in Figure 5, MagicDrive outperforms BEVGen and BEVControl in achieving this consistency, offering a more effective solution.
>
> ---
> ### Weaknesses 2: consistency across adjacent frames.
> It could be relatively easy to implement temporal consistency for video generation with MagicDrive. For instance, we applied the concept from 'tune-a-video' [1] to modify a low-resolution version of our MagicDrive, and the results are presented in both Appendix F and on our [anonymous website](https://magic-drive.github.io/). Since this extension is straightforward, it is not highlighted as a major contribution in our paper.
>
> ---
> ### Question 1: do you have different CFG weights for different conditions?
> No. We consider whether CFG for map condition exists with others or not, and different weights (scales) for CFG, as shown in Figure 7. If exist, they share the same CFG weight.
>
> ---
>
> [1] J. Z. Wu, Y. Ge, X. Wang, S. W. Lei, Y. Gu, Y. Shi, W. Hsu, Y. Shan, X. Qie, and M. Z. Shou. Tune-a-video: One-shot tuning of image diffusion models for text-to-video generation. ICCV, 2023.

---

> > ### Comment · Reviewer_jXFZ · 2023-11-23
> >
> > Thanks for the feedback! I will remain my rating due to novelty concern.

---

> > > ### Author Response · Authors · 2023-11-23
> > > **Response to Reviewer jXFZ**
> > >
> > > Dear Reviewer jXFZ,
> > >
> > > We are pleased that we have adequately addressed your concerns regarding temporal consistency and other minor issues.
> > >
> > > In regards to novelty, **MagicDrive clearly demonstrates superior innovation compared to ControlNet**. ControlNet only considers pixel-level conditions such as a depth map or canny edges. MagicDrive, on the other hand, is the first work demonstrating the effectiveness of an addictive encoder branch for conditions (i.e., BEV map) outside of this category. Moreover, our street-view generation necessitates a significantly larger number of and more complex conditions than ControlNet, as well as multi-view consistency, none of which were previously featured in ControlNet.
> > >
> > > As demonstrated in Figures 5&6 of our paper, as well as on our website, MagicDrive not only generates highly realistic street views with cross-view consistency, but also has the ability to manipulate street views under a variety of 3D conditions. Moreover, we present evidence that the data generated from controllable generation can enhance performance in perception tasks such as BEV segmentation and 3D object detection. None of these concepts have been explored in previous publications.
> > >
> > > We hope our response has adequately addressed your concerns. We appreciate the time and effort you have invested in reviewing our work.
> > >
> > > Best regards,
> > >
> > > Paper 153 Authors

---

> ### Comment · Reviewer_jXFZ · 2023-11-23
>
> Sorry, please do not ignore my comments. I don't agree with the argument that MagicDrive clearly demonstrates superior innovation compared to ControlNet. On one hand, I appreciate the delivered quality of this paper. But I still question the novelty since it is rather minor in my opinion on top of ControlNet framework with additional adaptation of conditions for one specific task.
>
> As for cross-view consistency, correct me if I misunderstood the result, isn't that clear that it is not consistent from the website demo (image sequence for continuous scenes)?

---

> > ### Author Response · Authors · 2023-11-23
> > **Response to Reviewer jXFZ**
> >
> > Dear Reviewer jXFZ,
> >
> > Thank you for your quick reply.
> >
> > As we mentioned earlier, street view generation demands more than just pixel-level conditions like ControlNet. As demonstrated in the first row of Table 4, ControlNet alone is insufficient to guide street-view generation accurately, let alone ensure cross-view consistency. Thus, the "additional adaptation of conditions for one specific task" entails significant effort, and our proposed encoding methods for different conditions are innovative in comparison with ControlNet.
> >
> > Looking back, Mask R-CNN "extends Faster R-CNN by adding a branch for predicting an object mask", while ViT shows "a pure transformer applied directly to sequences of image patches". These "additional adaptation" "for one specific task" is indeed novel. Moreover, given the fact that previous research works tried street view generation, while MagicDrive is the first to deliver superior quality and controllability, this "additional adaptation of conditions for one specific task" can be considered a significant contribution to the field.
> >
> > In reference to "cross-view consistency", our website's "Image sequences for continuous scenes" section is designed to showcase the diversity of our generation capabilities. This section demonstrates variations between frames without temporal consistency, while ensuring consistency across six different camera perspectives. For results related to temporal consistency (i.e., "consistency across adjacent frames", as discussed in Weakness 2), please refer to **the final part** of our website, specifically the **"Video Generation"** section and Appendix G. In these sections, we demonstrate how easily MagicDrive can be extended for video generation.
> >
> > We sincerely appreciate your time and effort in reviewing our work and engaging in this discussion.
> >
> > Best regards,
> >
> > Paper 153 Authors

---

> ### Comment · Reviewer_jXFZ · 2023-11-23
>
> Thanks for follow-up with additional clarification. With regards to "cross-view consistency", in [UPPER] Generated video 1 from MagicDrive. [LOWER] Ground truth for video 1, it is not consistent, isn't it? The street window changes across frames, right? I am not sure it makes sense to evaluate on "temporal consistency" for this task -- street view generation.
>
> Thanks again for author's detailed and clear clarification! My rating is final. Don't get me wrong, I do believe this is a great paper, but I kinda expect more on the claimed contribution.

---

> > ### Author Response · Authors · 2023-11-23
> > **Response to Reviewer jXFZ**
> >
> > Dear Reviewer jXFZ,
> >
> > We would like to emphasize that our primary contribution does not lie in video generation. We primarily focus on 3D geometry controllability in single frames and cross-view consistency for realistic street view generation. We are merely demonstrating the potential of MagicDrive to extend to video generation, as requested by the reviewers. We will continually enhance the fine-grained details in future work.
> >
> > Nonetheless, when compared with the results from other video generation works, such as [VideoLDM](https://research.nvidia.com/labs/toronto-ai/VideoLDM/) and [DriveDreamer](https://drivedreamer.github.io/), our preliminary results hold up well. The controllable generation and cross-view consistency provided by MagicDrive plays a crucial role in street view generation and downstream perception tasks.
> >
> > We appreciate the time and effort you have invested in reviewing and discussing our work. Your suggestions are important for improving this work.
> >
> >
> > Best regards,
> >
> > Paper 153 Authors

---

### Official Review · Reviewer_dbzg · 2023-11-01

**Soundness:** 3 good
**Presentation:** 3 good
**Contribution:** 2 fair
**Rating:** 5
**Confidence:** 4

**Summary:**

The paper proposes MagicDrive - a Bird's-eye-view(BEV)-to-street-view image generation method. Given a BEV road map, 3D bounding boxes for objects, the camera pose, and an input prompt it generates a consistent, multi-view image set for autonomous driving purposes. It is capable of scene, background and foreground control by prompting - lighting conditions, weather, object orientation, object deletion are available.

The main paper contributions are the a view-consistent image generation and 3D bounding box encoding for objects, as opposed to previous approaches that used only the BEV map.

The algorithm yields favorable visual results compared to similar methods (15-16 FID vs 20+) and the augmented data it generates improves upon the BEVFormer 3D object detection (\~+2mAP, depending on input modality) and Cross View Transformer vehicle and road mIoU (\~4-5%) on the nuScenes validation dataset.

**Strengths:**

- consistent cross-camera image generation
    - a cross-view attention model with neighboring views
- better problem modelling compared to older methods
    - 3D object bounding box and camera pose allows a wider array of edits and more accurate terrain representation
    - prompting the diffusion model allows for more diverse output images

**Weaknesses:**

## Summary ##
A view-consistent UNet generation method and bounding box inputs for a controlNet BEV-to-RGB are the main contributions. Apart from the benefit of encoding bounding boxes, unclear whether the chosen consistency method is ideal.
## Details ##

- engineering work / limited contributions
    - ControlNet stable diffusion pipeline coupled with a multi-view conditional UNet
        - there are other consistency methods - inpainting, panorama input, feature volumes - why is this cross-view attention module the best choice?
- limited comparisons, different baseline numbers
    - the authors use BEVFormer for some comparisons and CVT for others
        -  for BEVFormer the reported numbers are significantly lower compared to the original paper and I don't believe it's only the resolution; no numbers match
- method not mature enough
    - to the best of my knowledge, neither of the two baselines (BEVGen/BEVControl) have been accepted at a major conference; furthermore, MagicDrive disregards other practical considerations such as temporally-consistent frames [1*]
___
[1*] DrivingDiffusion: Layout-Guided multi-view driving scene video generation with latent diffusion model. arXiv preprint arXiv:2310.07771.

**Questions:**

1. Why are the BEVFusion numbers much lower? Why not use BEVFusion for the BEV segmentation as well?
2. If the aim is just to generate novel views, why not add additional elements to the bounding box images and use controlNet image encoding? See [1*] for inspiration.
3. If the data augmentation strategy works so well, why not start with a state-of-the art method such as [2*] and see what it can be improved from there?
4. Why not present other methods for consistent view generation? Arguably [4*] deals with the same problem; the scope is different, but they also have reasonable depth maps.
5. The method is heavily reliant on nuScenes; how would you consider improving generalization?

___
[1*]Li, X., Zhang, Y., & Ye, X. (2023). DrivingDiffusion: Layout-Guided multi-view driving scene video generation with latent diffusion model. arXiv preprint arXiv:2310.07771.
[2*] Hu, H., Wang, F., Su, J., Hu, L., Feng, T., Zhang, Z., & Zhang, W. EA-BEV: Edge-aware Bird’s-Eye-View Projector for 3D Object Detection.
[3*]Höllein, L., Cao, A., Owens, A., Johnson, J., & Nießner, M. (2023). Text2room: Extracting textured 3d meshes from 2d text-to-image models. arXiv preprint arXiv:2303.11989. https://github.com/lukasHoel/text2room
[4*]Bahmani, S., Park, J. J., Paschalidou, D., Yan, X., Wetzstein, G., Guibas, L., & Tagliasacchi, A. (2023). Cc3d: Layout-conditioned generation of compositional 3d scenes. arXiv preprint arXiv:2303.12074. https://github.com/sherwinbahmani/cc3d

---

> ### Author Response · Authors · 2023-11-16
> **Response to Reviewer dbzg (1/3)**
>
> Dear Reviewer dbzg,
>
> We thank you for highlighting our work's strength in "*better problem modeling compared to older methods.*" Indeed, this aspect is also one of the major novelties of our paper. Here, we present a point-by-point response to the weaknesses proposed by the reviewer.
>
> ---
> ### Weakness 1: novelty of the proposed method, especially the difference between ControlNet, and the design of multi-view consistency module.
>
> 1. **Differences between ControlNet and MagicDrive.**
>
>    **Firstly, our task diverges significantly from ControlNet's scope**. The complexity of street view generation mandates considering multiple, interdependent conditions (e.g., BEV map, 3D object parameters, and camera pose). For example, given bounding boxes with 3D coordinates, only if the camera pose is presented can determine which box to appear in the view; given maps in BEV, the height of terrain should be reasoning from the height of objects' positions. This multi-faceted approach transcends ControlNet's simpler image-space controls by necessitating intricate interactions and perspective-aware condition handling.
>
>    **Secondly, our approach distinctively extends ControlNet by integrating a diverse set of conditions.** Our conditions differ substantially from standard images. We demonstrate in Sec. 6 and Table 4 (see w/o $E_{box}$ example) that our tailored encoding mechanisms are crucial for achieving high-quality results, especially in maintaining multi-view consistency.
>
> 2. **Challenges and non-trivial design in achieving multi-view consistency.** Implementing cross-view consistency in street view generation presents unique challenges. Unlike indoor settings, each street view has limited overlap and varies significantly in camera characteristics (e.g., intrinsic properties, exposure), as exemplified in Figure 5. Our MagicDrive notably outperforms existing methods like BEVGen and BEVControl in achieving more effective consistency, as demonstrated in the comparative analysis in Figure 5.
>
> 3. **Reasons for choosing cross-view attention over alternatives.** Other consistency methods, such as inpainting or panorama inputs, typically assume significant overlaps and uniform view characteristics, which are not feasible in street view scenarios. Our cross-view attention module, being straightforward yet highly effective, addresses these limitations by accommodating the disparate nature of street views, thereby surpassing the performance of other baseline methods.
>
> ---
> ### Weakness 2: limited comparisons with baselines and different baseline numbers.
>
> 1. **Reasons for using BEVFusion\* and CVT as baselines.** In the evaluation, we consider CVT for BEV segmentation because it is used in both BEVGen and BEVControl. We consider BEVFusion because it is one of the most influential works in 3D BEV object detection, which is also referenced as one of the baselines in DrivingDiffusion [1].
>
> \* We think BEVFormer in the comments is a typo of BEVFusion.
>
> 2. **Why the reported numbers of BEVFusion are significantly lower compared to the original paper?** We have updated the number in Table 2 and Table 6 by enabling `test_mode` as described in [issue \#411](https://github.com/mit-han-lab/bevfusion/issues/411), which we did not notice when submitting our paper. Taking Table 6 "CAM-Only 2x" as an example, BEVFusion reports 35.56 mAP, while our reproduced mAP is 35.49. The mAP with data from MagicDrive is 35.74, improving the mAP by 0.25.

---

> ### Author Response · Authors · 2023-11-16
> **Response to Reviewer dbzg (2/3)**
>
> ### Weakness 3: the practicality of MagicDrive.
>
> 1. **Problem settings and practical applications.** Street-view data generation is a promising way to improve the perception capabilities of autonomous vehicles. The key challenges of this task include achieving controllability and ensuring multi-view consistency. Our approach addresses these problems effectively. Additionally, we utilize nuScenes, a widely recognized real-world dataset for autonomous driving, to demonstrate the realism of our generated street views. The practicality of our method is further substantiated by evaluating its performance on two different perception tasks, both in training and testing environments. This underscores the relevance and applicability of our settings to real-world scenarios.
>
> 2. **Reasons for choosing BEVGen and BEVControl as baselines.** At the time of our submission, these baselines represented the forefront of using controllable generation for street view synthesis, addressing the challenge of multi-view consistency. Our comparison demonstrates that MagicDrive significantly surpasses these baselines in generation quality. Moreover, our experiments using the CVT show substantial improvements over BEVGen, highlighting the advancement our approach brings to the field. We do not apply DrivingDiffusion as a baseline since it was released on arXiv after the ICLR submission deadline.
>
> 3. **Extending MagicDrive to achieve temporal consistency.** Ensuring temporal consistency in the generation of MagicDrive is a relatively straightforward process. To illustrate, we have implemented a technique inspired by the method from tune-a-video[2], applying it to fine-tune a low-resolution version of MagicDrive. The preliminary results of this enhancement are detailed in Appendix F, and a demonstrative video is available on our [anonymous website](https://magic-drive.github.io/). These demonstrate that MagicDrive is not only scalable but also serves as a robust foundational model for controllable street-view generation.
>
> ---
> ### Question 1: experiments of BEVFusion.
>
> 1. **Why are the BEVFusion numbers much lower?** Sorry for the confusion caused. We have updated the results in Table 2 and Table 6 by enabling `test_mode` as described in [issue \#411](https://github.com/mit-han-lab/bevfusion/issues/411) in the GitHub of BEVFunsion, which we did not notice when submitting our paper. The updated results are close to those reported in the paper of BEVFusion. Still, MagicDrive improves the performance of 3D object detection.
>
> 2. **Why not use BEVFusion for BEV segmentation?** In fact, we did employ BEVFusion for BEV segmentation. However, due to the absence of an established baseline for comparison, we initially refrained from presenting these results in our submission. To address this, we have now included the BEVFusion-tested results for BEV segmentation on the nuScenes validation set in Appendix F．
>
> ---
> ### Question 2: advantages of our methods compared with projecting 3D bounding boxes onto the image views like DrivingDiffusion.
> **First, for 3D object encoding**, as illustrated in Figure 2 of Sec. 1, it is crucial for accurate 3D control. After projection from 3D to 2D, the occlusion relationship and connection between views, for example, is unclear. Please see the visualization in Figure 13 for further understanding. It will be much clearer if 3D positions are available. **Second, for raw BEV maps**, as demonstrated in the last paragraph of Sec. 4.1, projecting a map from BEV to FPV without height information is an ill-posed problem. Therefore, there is no guarantee that this projection will not affect fidelity.
>
> ---
> ### Question 3: why not start with a SOTA perception method for data augmentation?
> Our choice of BEVFusion as the baseline is motivated by its significant influence and robust performance in the field. The primary focus of our method is to explore the impact of generating high-quality street-view data, rather than to establish new benchmarks in downstream perception tasks. The improvements we have demonstrated in these tasks serve as a proof of concept for the effectiveness of our approach. In terms of generation capabilities, our method surpasses existing baselines. Furthermore, while EA-BEV utilizes 1600x900 resolution images, replicating this in our training and generation processes is challenging due to limitations in resources and time.
>
> ---
> ### Question 4: why not present other methods for consistent view generation?
>
> Please refer to the response to Weakness 1, where we stated several key differences between street view generation and other settings that request view consistency. Additionally, we note that mentioned CC3D [3] offers street view generation on their website, but the quality is inferior compared to ours.

---

> ### Author Response · Authors · 2023-11-16
> **Response to Reviewer dbzg (3/3)**
>
> ### Question 5: reasons for choosing nuScenes and the generalization ability of our method.
> Note that nuScenes is a typical autonomous driving dataset featuring multiple diverse driving views. Therefore, we chose it as our testbed. Our method is based on stable diffusion, pre-trained with billions of images, demonstrating strong generalization ability in variant applications. Given a larger and more diverse dataset, we anticipate that MagicDrive would exhibit enhanced generalization in generating novel street views.
>
> ---
>
> [1] X. Li, Y. Zhang and X. Ye. DrivingDiffusion: Layout-Guided multi-view driving scene video generation with latent diffusion model. arXiv preprint arXiv:2310.07771, 2023.
>
> [2] J. Z. Wu, Y. Ge, X. Wang, S. W. Lei, Y. Gu, Y. Shi, W. Hsu, Y. Shan, X. Qie, and M. Z. Shou. Tune-a-video: One-shot tuning of image diffusion models for text-to-video generation. ICCV. 2023.
>
> [3] S. Bahmani, J. J. Park, D. Paschalidou, X. Yan, G. Wetzstein, L. Guibas and A. Tagliasacchi. CC3D: Layout-conditioned generation of compositional 3d scenes. arXiv preprint arXiv:2303.12074. 2023.

---

### Author Response · Authors · 2023-11-16
**Summary of Revision**

We thank all reviewers for their time, insightful suggestions, and valuable comments. Below, we respond to each reviewer’s comments in detail. We have also revised the main paper and the appendix according to the reviewers’ suggestions. The main changes are listed as follows:

1. We changed some demonstrations in Sec. 1 and Sec. 2 to better emphasize our novelty and contribution.
2. According to Reviewer TY6p, we added a footnote on the second page to restrict the scope of the term "Geometry Control".
3. We fixed the typo in Eq.9 identified by Reviewer TY6p.
4. In Table 1, we provide reference results on real data by downsampling them to $224\times 400$.
5. By enabling `test_mode` as stated in [issue #411](https://github.com/mit-han-lab/bevfusion/issues/411) on BEVFusion's GitHub, we updated the results in Tables 2 and 6 for BEVFusion.
6. We added Appendix E "Detailed Analysis on 3D Object Detection With Synthetic Data" as requested by Reviewer UU4E in W2.
7. We added Appendix F "More Results For BEV Segmentation" as requested by Reviewer dbzg in Q1.
8. We added Appendix G "Extension to Video Generation" as requested by Reviewer dbzg in W3, Reviewer UU4E in Q1, and Reviewer jXFZ in W2.
9. We added Appendix H "Generalization of Camera Parameters" in response to Reviewer TY6p in W2.

Note that we mark all the revisions in blue. We hope that our efforts can address the reviewers’ concerns well. We also updated our [anonymous website](https://magic-drive.github.io/) to present more results on generation diversity and video generation. Thank you very much again!


Best regards,

Paper 153 Authors

---

### Author Response · Authors · 2023-11-20
**Response to All Reviewers**

Dear Reviewers,

We greatly appreciate the time and expertise you have devoted to reviewing our work. Your insightful comments and critiques have been invaluable to us, and we have worked diligently to address your concerns in our response and the revised manuscript. We are eager to continue this fruitful discussion and are looking forward to your feedback. Thank you once again for your time and commitment.

Best regards,

Paper 153 Authors

---

### Author Response · Authors · 2023-11-23
**Request for Review Feedback**

Dear All Reviewers,

As the deadline for the review process draws near, we eagerly anticipate your valuable feedback. We understand the various responsibilities you handle and stand ready to address any outstanding queries regarding our draft.

Please do not hesitate to reach out if you require further information or clarification about our work.
Thank you in advance for your time and dedication.


Best Regards,

Paper 153 Authors

---

### Meta-Review · Area_Chair_YRHg · 2023-12-06

**Metareview:**

The paper targets at the street view scene generation. The paper proposes a framework that provides different 3D controls as conditions. Two reviewers recommend acceptance, while two reviewers rated it below the bar. Overall, reviewers agree that the problem setup and the visual results are good. Original concerns lied in the insufficient downstream use cases, video generation quality, and some ablation analysis. These concerns, however, I believe the author addressed most of them and make the experiments more thorough by providing substantial additional results in the rebuttal phase. The remaining concerns left lie in the scientific significance compared to previous methods, like Controlnet, and the visual consistency when shown in video format. These concerns d have their points but I believe the results of this work sufficiently provide values to the related community.

**Justification For Why Not Higher Score:**

The paper does have limitations in terms of technical contributions and corroborate its usefulness in downstream tasks.

**Justification For Why Not Lower Score:**

The paper provides thorough justifications to the concerns raised by reviewers. The additional comparisons, experiments, and visual results are convincing.

---

### Decision · Program_Chairs · 2024-01-16

Accept (poster)